# Global fossil fuel reduction pathways under different climate mitigation strategies and ambitions

Ploy Achakulwisut [1,2] ✉, Peter Erickson[1], Céline Guivarch [3], Roberto Schaeffer [4], Elina Brutschin [5] & Steve Pye [6]

The mitigation scenarios database of the Intergovernmental Panel on Climate Change's Sixth Assessment Report is an important resource for informing policymaking on energy transitions. However, there is a large variety of models, scenario designs, and resulting outputs. Here we analyse the scenarios consistent with limiting warming to 2 °C or below regarding the speed, trajectory, and feasibility of different fossil fuel reduction pathways. In scenarios limiting warming to 1.5 °C with no or limited overshoot, global coal, oil, and natural gas supply (intended for all uses) decline on average by 95%, 62%, and 42%, respectively, from 2020 to 2050, but the long-term role of gas is highly variable. Higher-gas pathways are enabled by higher carbon capture and storage (CCS) and carbon dioxide removal (CDR), but are likely associated with inadequate model representation of regional $CO_2$ storage capacity and technology adoption, diffusion, and path-dependencies. If CDR is constrained by limits derived from expert consensus, the respective modelled coal, oil, and gas reductions become 99%, 70%, and 84%. Our findings suggest the need to adopt unambiguous near- and long-term reduction benchmarks in coal, oil, and gas production and use alongside other climate mitigation targets.

The 2021 Glasgow Climate Pact reaffirmed the commitments of national governments to limit global-mean temperature rise to 1.5 °C[1], and recent developments in climate governance suggest increasing policy attention on fossil fuels in the years ahead. For example, in 2021, the 26th Conference of the Parties (COP26) to the United Nations Framework Convention on Climate Change (UNFCCC) saw a launch of the first-ever alliance of governments working together to "facilitate a managed phase-out of oil and gas production"[2]. COP26 culminated in an Agreement text that explicitly mentioned fossil fuels for the first time, with countries committing to accelerate efforts "towards the phasedown of unabated coal power"[1]. At COP27, more than 80 countries supported India's proposal to apply this phase-down language to all fossil fuels[3]. In March 2023, the UN Secretary-General urged all

countries to establish a "global phase-down of existing oil and gas production compatible with the 2050 global net zero targets"[4]. At the same time, global fossil fuel-derived carbon dioxide ($CO_2$) emissions rose to the highest level in history in 2021–2022[5], and the global levels of fossil fuel production being planned and projected by governments' national energy outlooks remain vastly in excess of levels consistent with limiting warming to 1.5 °C[6]. The 2022–2023 global energy crisis has also sparked a reappraisal of energy policies and priorities worldwide. It remains unclear whether this will serve to accelerate the clean energy transition or lock in fossil fuel dependence[7], with many countries expanding natural gas production or import capacity and others reviving coal use in their short-term responses[8]. This occurs against a backdrop of long-standing and growing tensions as to whether natural

[1]Stockholm Environment Institute, Seattle, WA, USA. [2]Stockholm Environment Institute, Bangkok, Thailand. [3]International Research Center on Environment and Development (CIRED), École des Pont, Nogent-sur-Marne, France. [4]Centre for Energy and Environmental Economics (CENERGIA), COPPE, Universidade Federal do Rio de Janeiro, Rio de Janeiro, Brazil. [5]International Institute for Applied Systems Analysis (IIASA), Laxenburg, Austria. [6]UCL Energy Institute, University College London, London, UK. ✉e-mail: ploy@sei.org

gas is a "bridge fuel" to a low-carbon future or a "bridge to nowhere"[9–14].

Consequently, although policymakers now broadly share the common goal of striving for a net-zero future, new debates are emerging around the speed and trajectory of the necessary energy transitions, including the associated reduction pathways for coal, oil, and natural gas (also known as fossil gas or fossil methane gas; hereafter referred to as gas) supply and demand[14–18]. Such debates are increasingly informed by process-based integrated assessment models (IAMs), which have become widely used to provide policy-relevant insights into how the world's energy and land-use systems can be transformed in the most cost-effective way to achieve the Paris Agreement's goal of limiting long-term warming to "well below 2 °C and pursuing efforts to limit the temperature increase to 1.5 °C above pre-industrial levels"[19,20].

To meet a given carbon budget – the cumulative amount of $CO_2$ emissions that can be emitted until the end of the century or until net-zero attainment – consistent with limiting global warming to a certain temperature threshold, IAMs generally rely on different combinations and extents of the following major strategies: (1) phasing out fossil fuels from the energy supply, buildings, transport, and industry sectors; (2) transforming agricultural and land-use practices; (3) reducing energy and material consumption; and (4) relying on carbon dioxide removal (CDR) and carbon capture and storage (CCS) deployment. Non-$CO_2$ greenhouse gases (GHGs) such as methane and other short-lived climate forcers also influence the level of peak warming, and their emissions reductions are another important mitigation lever[21]. The relative importance of different mitigation strategies reflects differences in the underlying model framework, scenario design, and input parameters and assumptions such as technological performance and adoption, economic relationships, and cost optimization[22–24]. The majority of low-carbon scenarios produced by IAMs rely extensively on CDR, mostly through bioenergy combined with carbon capture and storage (BECCS) and land sequestration; a few scenarios also employ direct air capture with carbon capture and storage (DACCS)[21,25]. However, the feasibility of large-scale CDR and CCS deployment remains highly uncertain, with growing concerns that we would be locking in continued fossil fuel dependence and global temperature increases above 2 °C if they were to fail[26–30]. Intensive land use for bioenergy or for carbon sequestration by the Agriculture, Forestry, and Other Land Use (AFOLU) sector could also lead to land degradation, food insecurity, biodiversity loss, and water scarcity[28,29,31].

The Working Group III (WGIII) contribution to the Intergovernmental Panel on Climate Change (IPCC)'s Sixth Assessment Report (AR6) compiled and assessed 3131 scenarios generated by almost 100 different model versions from more than 50 model families, with varying regional scope and temperature outcomes[21]. This new scenarios database is likely to be extensively used to inform socio-political discourse on energy system transformations in the years ahead, including on identifying fossil fuel reduction pathways that are consistent with the Paris Agreement[32,33]. However, there is a large variety of IAMs and associated modelling approaches and assumptions, resulting in important differences in the outputs. Moreover, there is an increasing risk that key assumptions and the resulting model-based insights become implicitly accepted by researchers and policymakers without a clear understanding or critical evaluation of their real-world implications, feasibility, and desirability[34].

In this work, we explore what the AR6-assessed scenarios consistent with limiting warming to 2 °C or below say about the speed, trajectory, and feasibility of different global coal, oil, and gas reduction pathways and why. We first summarise the trends across all modelled pathways under three different temperature outcomes, and perform a classification and regression tree analysis to identify the most important factors influencing the levels of cumulative supply for a given fossil fuel. We then perform additional statistical analyses to further

explore the model parameters and assumptions associated with different transition pathways for gas. Next, we evaluate the sensitivity of the modelled fossil fuel reduction trajectories to different mitigation scenario storylines and assumptions, including CDR reliance, and their feasibility with respect to geophysical, economic, technological, socio-cultural, and institutional considerations. We then close with a discussion of the policy implications of our findings.

## Results
### Global fossil fuel reduction pathways in scenarios that likely limit warming to 2 °C or below
Chapter 3 of the IPCC AR6 WGIII report vetted 1686 global scenarios, of which 1202 provided sufficient information for a temperature outcome categorization, ranging from C1 to C8[21]. Here we focus on the categories with the three lowest temperature outcomes: C1 – limit warming to 1.5 °C in 2100 with a likelihood greater than 50%, with no or limited overshoot; C2 – limit warming to 1.5 °C in 2100 with a likelihood greater than 50%, with high overshoot; and C3 – limit peak warming to 2 °C with a likelihood greater than 67%[21]. We analyse all scenarios in the C1-C3 ensemble and consider them to be relevant to the Paris Agreement's temperature limits, even if some individual scenarios may be considered to not be fully Paris-compliant depending on one's interpretation of its long-term temperature goal and of its other objectives, as well as judgement on the probability, overshoot, and timing of the temperature change[35]. Given the Glasgow Climate Pact's emphasis on the 1.5 °C limit, we pay particular attention to the C1 scenarios in the main text. Of the 541 vetted C1-C3 scenarios, 94 C1, 131 C2, and 310 C3 scenarios report primary energy supply from coal, oil, and gas. Since all but one C3 scenario analysed in this study are generated from IAMs that typically include non-energy uses of fossil fuels under their reporting of the "Primary Energy|xx" variable (see Methods for details), we broadly interpret this variable to represent total supply intended for all uses. However, the completeness with which non-energy uses are accounted for and reported varies between different models and could have significant implications for the resulting fossil fuel reduction pathways under a given carbon budget. Non-energy uses can lead to either long-term carbon storage in stable physical products or eventual combustion (e.g., incineration of discarded plastics), with historical estimates suggesting that around 0.02% of coal, 8.02% of oil, and 1.86% of gas produced do not lead to eventual carbon emissions[36].

Figure 1 shows the individual, median, and interquartile range (IQR) 2010–2100 pathways, as well as boxplot distributions of the cumulative 2020–2100 values, of the global coal, oil, and gas supply as modelled by the C1-C3 scenarios (see Supplementary Figs. 1 and 2 for boxplot distributions of the annual values at certain years and of the peak years of supply, respectively). Showing the median pathway provides one way to succinctly communicate the average trajectory within a given scenario ensemble. However, this approach has limitations[37], especially because the AR6 ensemble is unstructured and does not represent a statistical sample: it merely reflects the options that have been explored in the underlying studies, which are represented by different model families and scenario designs and protocols to varying degrees (Supplementary Fig. 3). Moreover, while the modelled pathways of different mitigation options within a given scenario are internally consistent for meeting a certain carbon budget, the median pathways of different variables calculated from a set of scenarios may not be. Given this and the diversity in the model outputs, the median pathways shown in Fig. 1 should be considered for illustrative purposes only, as we will later explore how subsets of or individual scenarios diverge from them.

As shown in Fig. 1, across the C1-C3 scenarios, the global supply of coal and oil generally decline substantially and rapidly between now and mid-century, followed by a more gradual and variable reduction over time. More stringent temperature limits necessitate faster and

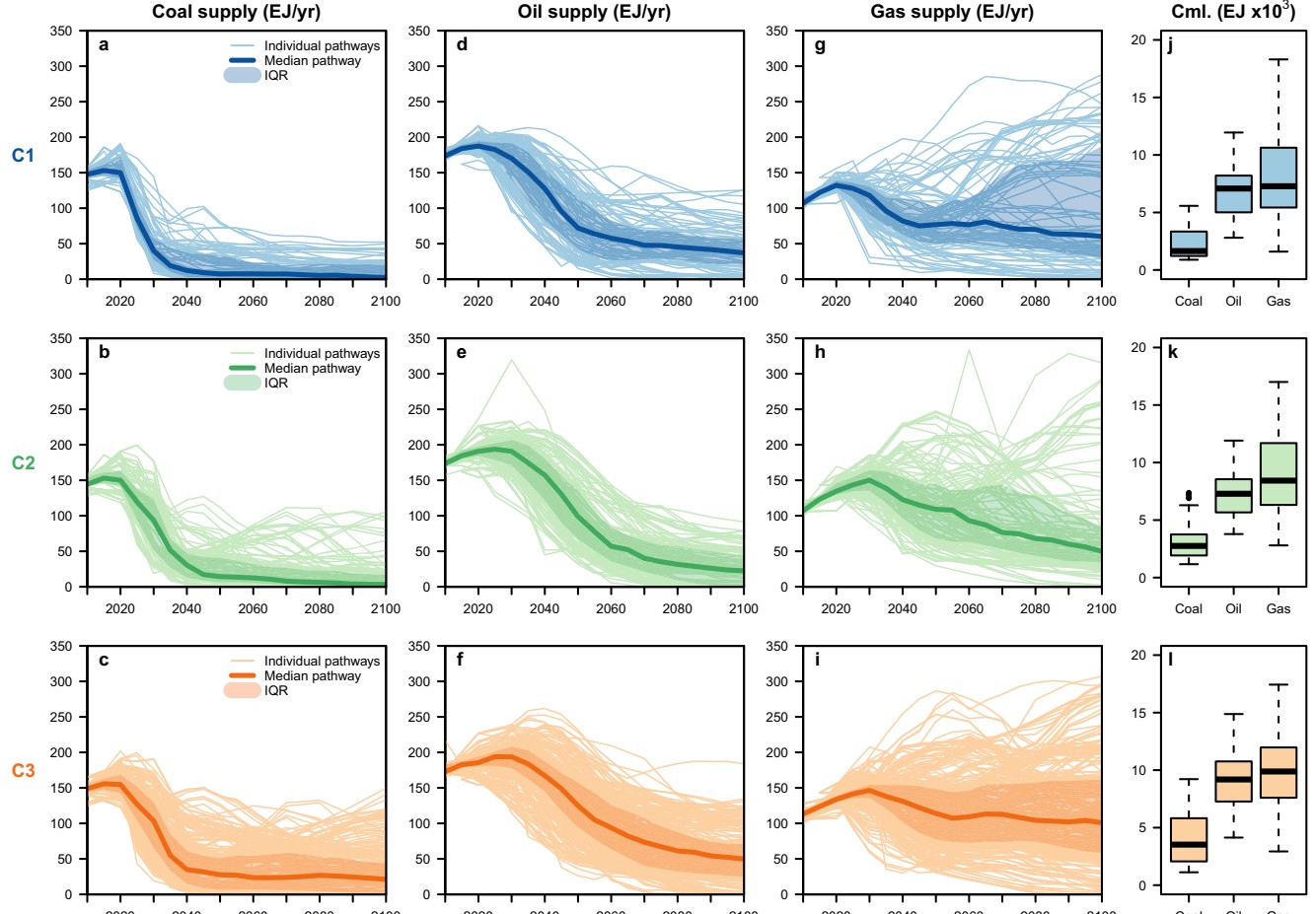

**Fig. 1 | Global primary energy supply (in exajoules, EJ) from coal, oil, and gas as modelled by the AR6-assessed mitigation scenarios consistent with limiting warming to 2 °C or below.** In panels **a**–**i**, the 2010–2100 annual timeseries of the following are plotted at 5-year intervals within a given temperature category (C1-C3): individual pathways (light lines); median pathways (dark lines); and the interquartile range (IQR, shadings). Panels **j**–**l** show the boxplot distributions of the 2020–2100 cumulative primary energy supply from coal, oil, and gas across the scenario ensemble within each temperature category. The horizontal center line depicts the median, the box spans the IQR between the 25th percentile (Q1) and 75th percentile (Q3), the lower whisker represents the minimum value or Q1 − 1.5 x IQR (whichever is larger), and the upper whisker represents the maximum value or Q3 + 1.5 x IQR (whichever is smaller).

greater reductions. Compared to coal and oil, there is less scenario consensus for the role of gas, with some seeing an almost complete phase-out by around 2050 while others see continued or increasing supply out to 2100. The scenarios also model varying levels of coal, oil, and gas supply that can be coupled to CCS at the point of combustion (Supplementary Figs. 4 and 5). Because there is such a large diversity in the pathways, we perform a classification and regression tree analysis (CART)[38] to identify, to first order, the most important factors that lead to different levels of cumulative 2020–2100 supply for a given fossil fuel. We choose to employ CART here due to its ability to combine statistical rigour with an accessible and visual interpretation of the results. The independent variables selected for inclusion are those that are frequently discussed in the literature[21,23,39] and/or judged to likely influence the level of fossil fuel dependence (see Methods). The CART results are shown in Fig. 2 (C1) and Supplementary Figs. 6–11 (C2-C3).

Across the C1-C3 scenarios, global coal supply on average peaks around 2015 (Supplementary Fig. 2) and rapidly declines (the fastest of all three fuels), with the median pathway reaching values below 10 exajoules per year (EJ/yr) in the C1 scenarios, and below 35 EJ/yr in the C2-C3 scenarios, after 2040 (Fig. 1a–c). Coal without CCS is largely eliminated by 2035–2040 (Supplementary Fig. 4). CART reveals that C1-C3 scenarios allowing relatively higher levels of cumulative coal supply are associated with certain model families and higher levels of coal coupled to CCS (Fig. 2a and Supplementary Figs. 6 and 7). For

example, six C1 scenarios with the highest levels of cumulative coal supply (around 5000 EJ) are associated with the AIM and GCAM model families and with cumulative coal coupled to CCS greater than or equal to 2650 EJ.

For oil, global supply on average peaks around 2020 for the C1 scenarios and around 2030 for the C2-C3 scenarios (Supplementary Fig. 2) followed by a steep decline out to mid-century, slower than coal but faster than gas (Fig. 1d–f). In general, C1 scenarios with the highest levels of cumulative oil supply are associated with higher oil demand for transportation and non-energy industry use and with higher coal supply; those with lower oil supply are associated with lower transportation demand, higher nuclear energy supply, and certain model families (Fig. 2b). For the C2 and C3 scenarios, higher oil levels are associated with certain model families (AIM, COFFEE, GCAM, REMIND, TIAM, and WITCH), higher gas supply, higher levels of fossil fuel use coupled to CCS ("fossil-CCS"), and certain scenario projects (Supplementary Figs. 8 and 9).

Among the three fuels, the pathways for gas display the largest variability. Nevertheless, global gas supply on average peaks around 2025, 2030, and 2035 for the C1, C2, and C3 scenarios, respectively (Supplementary Fig. 2). In the majority of scenarios, gas supply not coupled to CCS at the point of combustion peaks in 2020 and follows a similar decline trajectory to that of oil (Supplementary Fig. 4), while gas coupled to CCS generally increases out to mid-century

a) CART decision tree for 2020-2100 cumulative coal supply in C1 scenarios

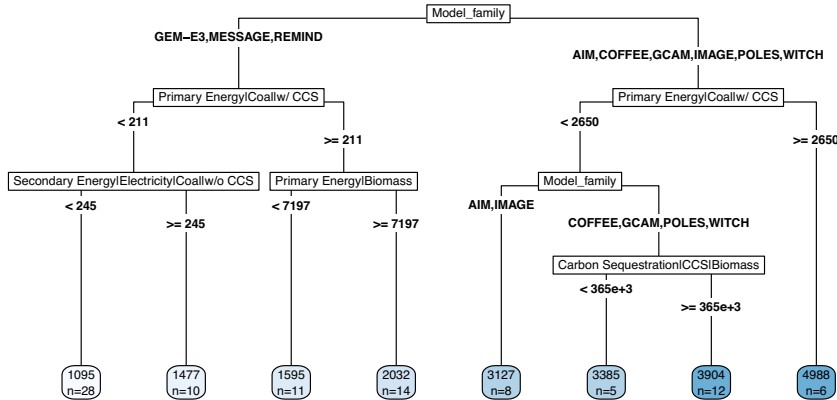

b) CART decision tree for 2020-2100 cumulative oil supply in C1 scenarios

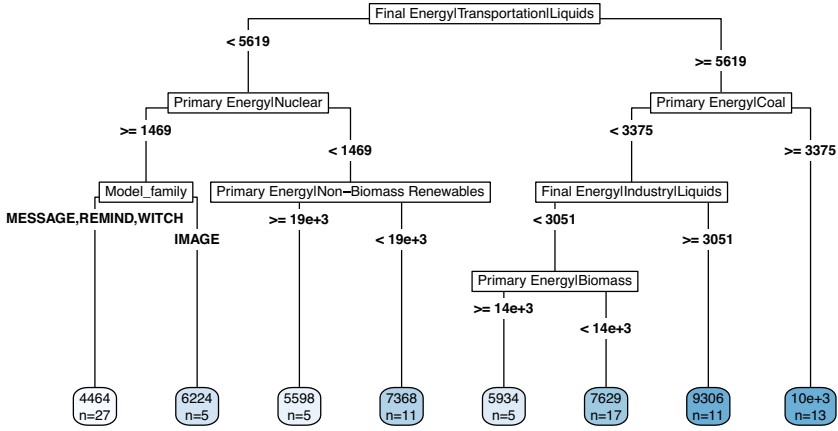

c) CART decision tree for 2020-2100 cumulative gas supply in C1 scenarios

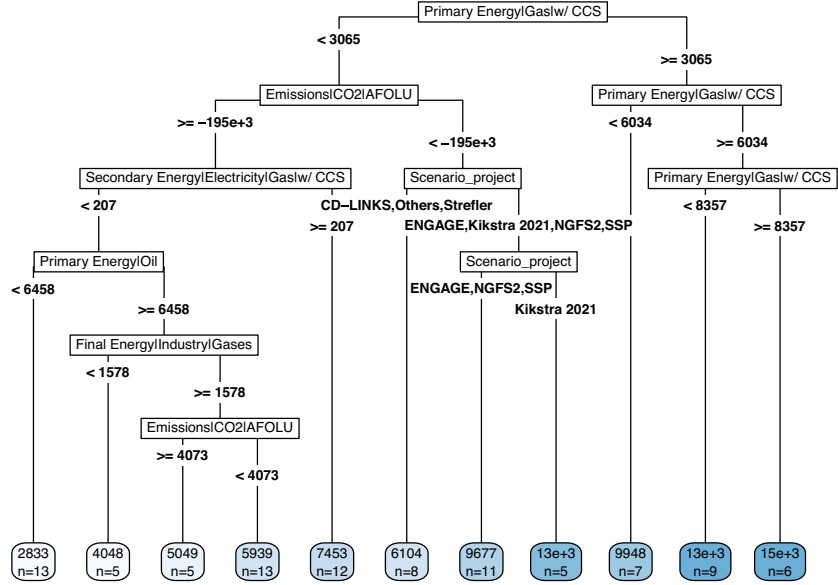

**Fig. 2 | Classification and regression tree (CART) analysis results for predicting the 2020–2100 cumulative fossil fuel supply in the C1 scenarios.** In panels **a–c**, the blue boxes at the bottom show the average value of cumulative coal, oil, and gas supply (in exajoules), respectively, across the the scenarios included within each given "leaf node" (note that they do not necessarily appear in increasing order from left to right). The units of the independent variables shown are exajoules for primary and secondary energy, and million tonnes of $CO_2$ for emissions and carbon sequestration (negative AFOLU emissions represent $CO_2$ removal). Cumulative 2020–2100 values are also used for the independent variables. (The root mean square errors (RMSE) of panels **a–c** are, respectively, 233, 734, and 928 EJ).

(Supplementary Fig. 5). Scenarios with relatively higher cumulative gas supply are consistently associated with higher levels of fossil-CCS across the C1-C3 scenarios (Fig. 2c and Supplementary Figs. 10 and 11). For example, six C1 scenarios with the highest cumulative gas supply (around 15,000 EJ) assume that 53%-70% of this supply can be coupled to CCS at the point of combustion. For C3, certain model families and scenario projects are also important predictors (Supplementary Fig. 11).

Since the model family is an important predictor of cumulative fossil fuel supply but is not equally represented in the AR6 ensemble (Supplementary Fig. 3), we test the sensitivity of the median pathways shown in Fig. 1 to potential model bias. There is still limited practice in and consensus on bias correction for model over-/under-representation of climate mitigation scenario ensembles[40]. Nevertheless, when we include only the scenarios with minimum and maximum 2020–2100 cumulative supply from each model family for a given fuel and category, the values and trajectories of the median pathways shown in Fig. 1 do not change much, except for C1-gas in which the post-2050 values would be higher (Supplementary Fig. 12).

Given the substantial variability in modelled gas supply, especially from 2050 onwards, and highly contested debates around the role of gas in energy transitions as previously discussed, we explore the model characteristics and assumptions that underly this variability in more detail in the next section.

## Characteristics of scenarios with different roles for gas

We first conduct an analysis of variance (ANOVA) to explore the influence of eight potential drivers of the variability in modelled gas supply over time between 2020–2100 (see Methods). Figure 3 shows that in the C1 scenarios, close to 75% of the variations in the modelled gas supply after 2040 can be explained by differences in the modelling frameworks and scenario designs. Differences in the amounts of fossil-CCS and gas intended for industry use (of which a large fraction goes towards non-energy applications such as for feedstocks, and this variable is also highly correlated with gas demand for transportation) also contribute. In the C2 and C3 scenarios, these four drivers are also the most important, with fossil-CCS becoming the most dominant factor towards the end of the century.

To identify some of the key common underlying dynamics and features of different scenario designs and modelling frameworks that lead to different roles for gas, we next perform a cluster analysis to divide the gas supply pathways into three different typologies according to their modelled values in the years 2030, 2050, 2075, and

2100 (see Methods). As can be seen in Fig. 4 and Supplementary Fig. 13, there are broadly three types of gas trajectories in the C1 and C2 scenarios: (1) rapid decline between now and mid-century followed by a more gradual reduction ("fast decline"); (2) a more gradual decline over time ("slow decline"); and (3) near-term decline followed by an increase in gas supply after around mid-century ("rebound"). In the C3 scenarios (Supplementary Fig. 14), we see the following three types of trajectories: (1) rapid decline between now and mid-century followed by a more gradual reduction ("decline"); (2) near-term decline followed by an increase in gas supply after around mid-century ("rebound"); and (3) increase to around mid-century followed by a plateau ("increase"). For each temperature category, Fig. 4 and Supplementary Figs. 13 and 14 also show the median and interquartile pathways of select model variables in each of the three gas clusters. The model family and scenario project distributions across the gas clusters are shown in Supplementary Figs. 15 and 16.

We summarize key findings across the C1–C3 scenarios here and provide a more detailed discussion in the Supplementary Information. The higher-gas "rebound" and "increase" C1-C3 scenarios are generally associated with one or more of the following features: (1) much higher levels of fossil fuel use coupled to CCS (panel i), including for gas-powered electricity generation (panel t); (2) higher CDR via negative AFOLU emissions (panel l) and DACCS (panel k); (3) lower carbon prices, especially after mid-century (panel n); (4) higher gas demand in the industry and transportation sectors (panels u-v); (5) higher primary energy supply from nuclear (C1 only; panel h); (6) lower capacity additions for electricity generation from solar, wind, and battery storage sources but higher for gas coupled to CCS (panels o-r); (7) lower capital and operation and maintenance costs for electricity generation from gas-CCS power plants in the near-term but higher from offshore wind across all years (panels y, ab, ac, and ae). (For the last two features, we interpret the relevant variables with caution given the relatively limited reporting.) Conversely, the "fast decline" C1–C2 scenarios are typically associated with much less reliance on fossil-CCS, higher carbon prices, and sustained increases in renewable capacity additions from 2020 onwards. The "slow decline" C1–C2 scenarios also share these characteristics, but generally have higher CDR via BECCS and less extremely high carbon prices seen in some of the "fast decline" scenarios. The reduction pathways for gas-powered electricity generation without CCS are, however, similar between the three clusters, showing an almost complete phase-out by around 2040–2060 across the C1–C3 scenarios (panel s).

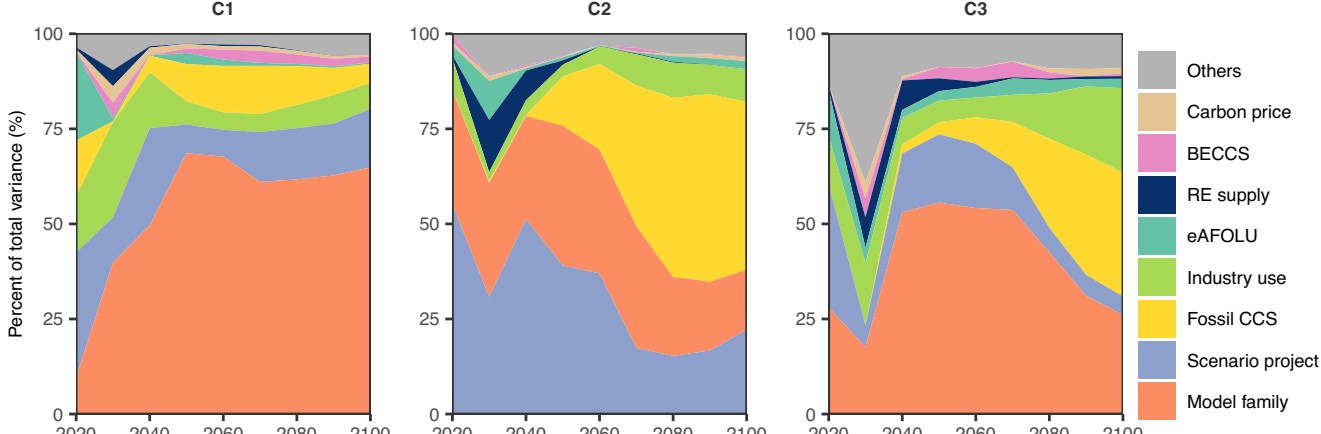

**Fig. 3 | Percent contributions of each driver to the variability in modelled gas supply over time between 2020–2100 for the C1-C3 scenarios.** A different colour is used for each driver. In each panel, the drivers are ranked in order of decreasing relative contribution across all time, from bottom to top. ("eAFOLU" = CO₂ sequestration by the Agriculture, Forestry, and Other Land Use sector; "RE supply" = primary energy supply from non-biomass renewables; "Others" = remaining, unexplained variance, which can include interactions between the drivers shown.) The "industry use" variable includes both fossil- and bio-based gases.

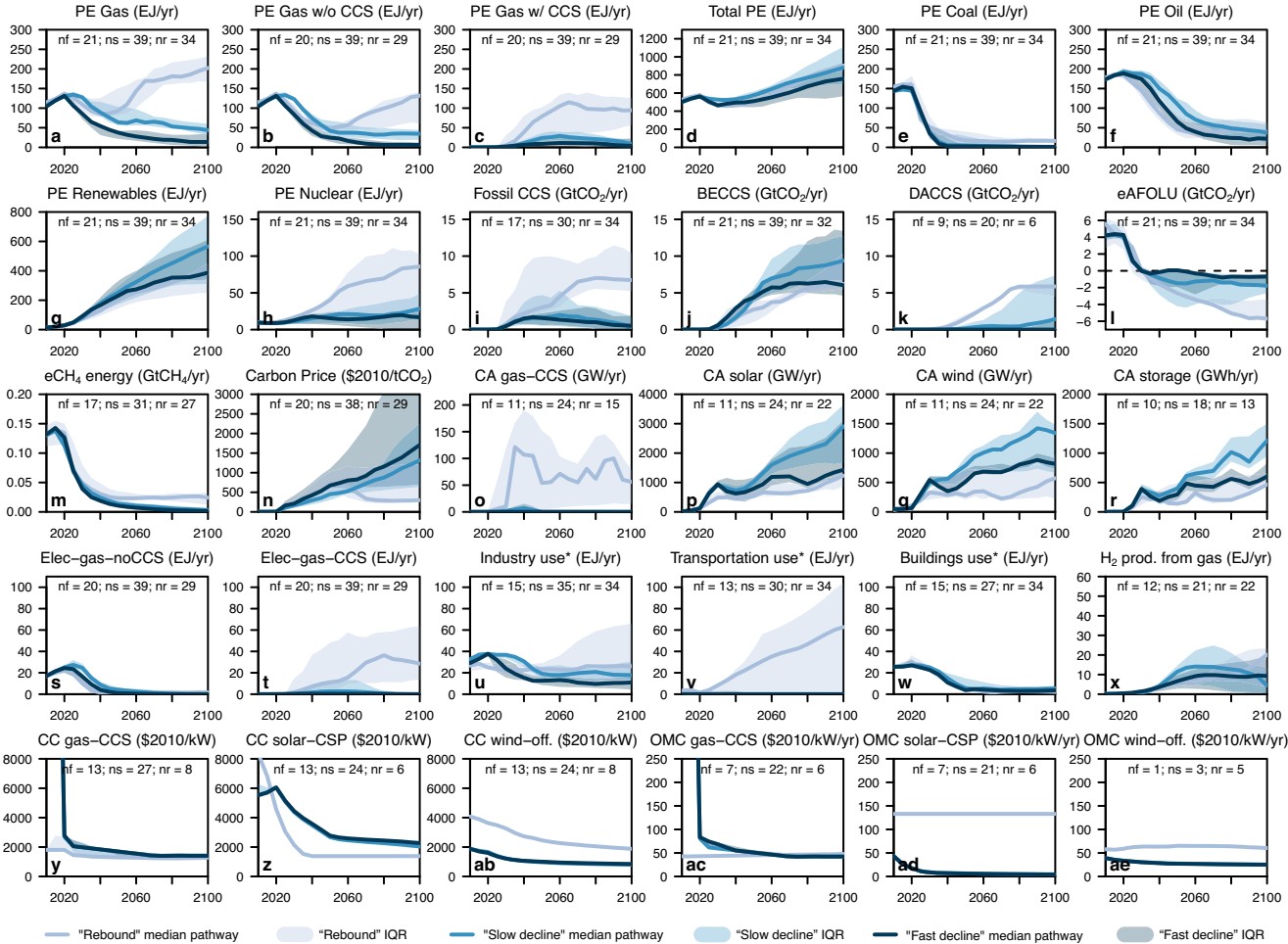

**Fig. 4 | 2010–2100 global pathways for select variables as modelled by the C1 scenarios divided into three clusters based on their 2030, 2050, 2075, and 2100 gas supply values.** The median (dark lines) pathway and interquartile range (shadings) of each cluster are plotted at 5-year intervals. The variables shown in each panel are as follows: **a** Primary energy from gas; **b, c** Primary energy from gas without and with coupling to carbon capture and storage (CCS); **d** Total primary energy supply; **e–h** Primary energy from coal, oil, non-biomass renewables, and nuclear; **i–k** CCS coupled to fossil fuel use, bioenergy, and direct air capture; **l** Carbon removal and sequestration by the Agriculture, Forestry, and Other Land Use sector; **m** Methane emissions from the energy sector; **n** Carbon Price; **o–r** Capacity additions to electricity generation from gas with CCS, solar, wind, and battery storage; **s–t** Electricity generation from gas without and with CCS; **u–w** Gas demand (from fossil- and non-fossil sources) for industry, transportation, and buildings; **x** Gas demand for hydrogen production; **y–ae** Capital costs (CC) and operation and maintenance costs (OMC) for gas with CCS, concentrated solar power, and offshore wind for electricity generation. (Notes: Variable units are shown inset; "nx" refers to the number of scenarios within each cluster that outputs a given variable, where x denotes the first letter of the cluster name).

Based on a combination of our ANOVA and gas-cluster analyses, an exploration of individual pathways for different gas clusters grouped by model families (Supplementary Figs. 17–19), and information drawn from the literature on relevant IAM features[23,24,30,41], we conclude that the gas trajectories in C1-C3 scenarios are influenced by one or more of the following factors, which are inter-dependent and tend to vary by model family: (1) carbon pricing; (2) constraints on the availability of CCS, CDR, and nuclear technologies; and (3) constraints on renewable deployment (such as assumptions on capital cost, operation and maintenance cost, levelized cost of electricity, and learning rates), resource availability, and integration challenges (e.g., assumptions about flexibility provision, grid expansion, and storage capacity). For example, REMIND has some of the highest carbon prices (Supplementary Fig. 17); imposes the most stringent assumptions on the global and regional $CO_2$ storage potentials and on the $CO_2$ injection rate (Supplementary Table 4); and assumes some of the lowest levelized costs of electricity from wind and solar compared to other models (see Fig. 3 in Luderer et al.[24]). Consequently, all but one C1 scenario from REMIND are grouped into the "fast decline" and "slow decline" clusters, with pathways showing some of the lowest levels of

fossil-CCS (Supplementary Fig. 18), and sustained increases in renewable capacity additions but very limited gas-CCS capacity additions, from 2020 onwards (Supplementary Fig. 19). These findings are consistent with those from Giannousakis et al.[41], who showed that the economics and depths of decarbonization modelled by REMIND are highly sensitive to transport sector costs, CCS injection rate assumptions, and renewable costs.

Conversely, all seven C1 scenarios from IMAGE model high gas supply, generally increasing until around 2060 followed by gradual declines (Supplementary Fig. 17). IMAGE has relatively low carbon prices, no higher than $USD_{2010}$ 1208 per tonne of $CO_2$ throughout the century, and imposes no constraints on the regional storage $CO_2$ potential or the injection rate (Supplementary Table 4). As a result, it models the highest levels of fossil-CCS (>10 GtCO₂/yr by mid-century) and gas-CCS capacity additions (Supplementary Figs. 18 and 19). Meanwhile, the majority of C1 scenarios from the MESSAGE model family show a gas revival after a near-term decline, displaying some of the highest levels of gas demand for the transport and industry sectors, enabled by high levels of CDR via AFOLU and BECCS and of fossil-CCS. MESSAGE does not impose any constraints on CCS availability

(Supplementary Table 4). This gas revival pattern, also seen in some of the scenarios from the GEM-E3 and WITCH models, may be partly driven by an unexplored but known model outcome associated with the "net-zero-budget" scenario design from the ENGAGE project[42], in which carbon prices initially increase but then stabilize or decline after net-zero $CO_2$ emissions are reached around mid-century and mitigation efforts are relaxed (Supplementary Fig. 17; also see Figure SI 1.1-6 in Riahi et al.[42]).

These differences between higher- and lower-gas scenarios help illustrate the diverging storylines and economics of how the energy transition can unfold to meet climate goals. In many of the higher-gas C1 scenarios, gas use declines rapidly through 2040 (Fig. 4a) but then increases rapidly again through the end of the century to levels far higher than the present. Such scenarios tend to front-load their ambition on oil and gas reductions in the 2030s and 2040s, only to see the deployment of CCS, nuclear energy, and CDR open up room in the carbon budget later in the century (and with corresponding reduced carbon prices) that re-invigorates (in the case of gas power) or newly invigorates (in the case of compressed natural gas-fueled transport vehicles) gas use. By contrast, the "fast decline" scenarios see dramatically faster rates of solar and wind power deployment, with flexible storage capacity that obviates the need for new gas power plants (even with CCS), and helps power the transport sector with more electricity than fossil- or bio-based gases.

How these two scenario typologies would evolve could also have dramatically different real-world implications for the development of technology innovation systems and the accompanying politics[43]. Nevertheless, given technology path dependencies (a common phenomenon in socioeconomic systems, which arises when initial conditions and their historical antecedents matter for eventual outcomes)[44], combined with the urgent need to rapidly decarbonize our energy systems, a gas phase-down followed by a revival seems questionable and calls for a careful justification that we could not find in the underlying studies. Due to the high heterogeneity in and limited transparency around IAM assumptions, combined with inconsistencies in variable reporting, we could not determine which model parameters and assumptions are specifically leading to this outcome. We offer three most likely possible explanations: (1) inadequate model representation of real-world constraints on fossil fuel- and renewable-based technology innovation, adoption, diffusion, and phase-in/-out path dependencies[44,45] (for example, an overestimation of the costs of renewable technologies is a problem that has been found in many IAMs[46,47] and which leads to higher mitigation value of CCS in electricity and hydrogen production[48]); (2) overly optimistic assumptions or insufficient constraints on CCS and CDR potential (for example, only the REMIND model imposes constraints on the regional $CO_2$ storage potential and injection rate, which influence fossil-CCS, BECCS, and DACCS availability)[30,49]; and, in the case of "net-zero-budget" scenarios specifically, (3) an unexplored but known model outcome in which carbon prices and mitigation efforts are relaxed after net-zero $CO_2$ emissions are reached in some of the models. In light of the important policy implications, we urge the model-scenario developers to more consistently report key model parameters, assumptions, and outputs, and to critically examine the real-world implications of their modelled results for gas pathways.

**Sensitivity of global fossil fuel reduction pathways to different mitigation strategies and uncertainties in CDR potential**
To further assess and demonstrate how much the reduction pathways of coal, oil, and gas are influenced by different scenario storylines and assumptions, in this section, we first focus on a number of individual illustrative scenarios and then explore what happens when we take a conservative approach to future CDR and fossil-CCS potential. The AR6 scenarios database identified five different "illustrative mitigation pathways" (IMPs) that reflect different prominent mitigation strategies

for reaching a given temperature outcome (see Fig. 5 legend). We emphasize that no two scenarios are alike or more "correct" than others; the IMPs are not representative of the AR6 ensemble. The individual coal, oil, and gas supply pathways under the five IMPs, along with the median pathways across each of the C1-C3 scenarios, are shown in Fig. 5. (Other select model variables from these pathways are also shown in Supplementary Figs. 20–22) We describe the fossil fuel decline pathways under the five IMPs in detail in the Supplementary Information, and highlight some key features of the IMP-LD, -SP, and -Neg scenarios here.

In the C1 category, the IMP-LD (low-energy demand) scenario charts out much faster reductions in oil and gas than almost all other scenarios (Fig. 5b, c). For oil, this is accomplished through widespread transitions to shared vehicle fleets, flexible transit systems, and rapid vehicle electrification, as well as reductions in freight volumes due to longer-lasting and more material-efficient goods. For gas, this is accomplished through extensive end-use efficiency improvements (e.g., through building retrofits), plus preferring renewable power to gas with CCS. In fact, the IMP-LD scenario deploys no fossil-CCS or BECCS for normative reasons, informed by concern over innovation failure, investment risks, and public opposition (though it does rely extensively on CDR via AFOLU; see Supplementary Fig. 20)[50].

In contrast, the IMP-SP's explicit focus on increasing access to affordable modern energy services (e.g., electric and LPG cook stoves) in developing regions causes it to reduce oil and gas demand slightly slower than the IMP-LD. IMP-SP subsequently phases out all fossil fuels almost completely by the end of the century and with relatively limited CDR and CCS reliance (Supplementary Fig. 20), driven in part by the "substantially reduce[d] detrimental effects of outdoor air pollution on public health", and by the potential negative environmental and social impacts of intensive land use-based CDR[51].

In the C2 category, the IMP-Neg scenario allows dramatically more room for coal than most scenarios but, over time, less room for gas (Fig. 5). The continued use of coal in this scenario is driven by industry demand, which is enabled by coupling to CCS and also by extensive CDR reliance: up to 8 $GtCO_2$ of BECCS, 1-2 $GtCO_2$ of sequestration by the AFOLU sector, and 6 $GtCO_2$ by other sequestration methods annually in the latter half of the century[42] (Supplementary Fig. 21). This extensive reliance on long-term CDR is typical of C2 scenarios, in which net negative $CO_2$ emissions are needed to compensate for emissions in the first half of the century and bring down temperatures after the peak[21].

Figure 5 also shows the median and interquartile ranges of C1-C3 pathways constrained by the potential 2020-2100 cumulative availability of BECCS (196 $GtCO_2$), afforestation (224 $GtCO_2$), and DACCS (320 $GtCO_2$) based on a recent expert consensus survey[49]. We find that 15%, 0%, and 10% of the C1, C2, and C3 scenarios, respectively, are within these limits (not all scenarios report the variables needed for this analysis; see Methods for details). The selected scenarios also have levels of total CCS (coupled to fossil fuel use, bioenergy use, and direct air capture) below 8.6 $GtCO_2$ per year around mid-century, which is the "investable" $CO_2$ storage capacity estimated by Grant et al.[30] after accounting for real-world regional differences in storage capacity and injection rates. Imposing these CDR limits implies much faster and greater reductions in fossil fuel supply and use compared to the full AR6 ensemble, especially for gas, in the near- and long-term. As Fig. 5 shows, the resulting median coal, oil, and gas reduction pathways fall around or below the 25th percentile of all scenarios. For example, under the median pathway calculated from all C1 scenarios, global coal, oil, and gas supply declined by 95%, 62%, and 42%, respectively, between 2020 and 2050. Under the median pathway of the CDR-limited scenarios, the respective declines are 99%, 70%, and 84% (see Supplementary Tables 1, 2 for details).

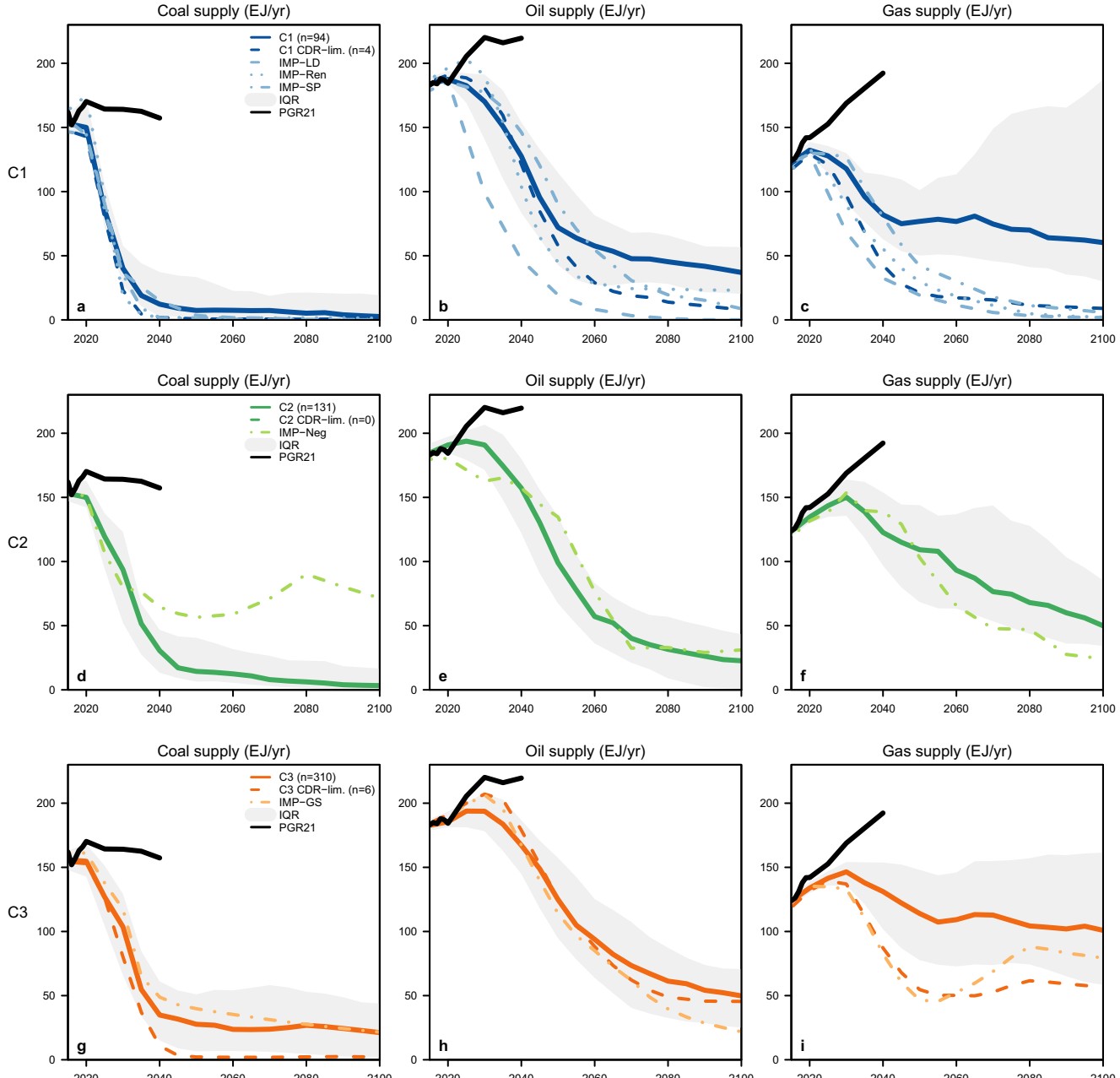

**Fig. 5 | 2015–2100 global pathways of fossil fuel supply under different individual or subsets of AR6-assessed scenarios and compared to those based on government plans and projections.** Panels **a**–**i** show global pathways of primary energy supply (in exajoules, EJ) from coal, oil, and gas under different subsets of or individual C1–C3 pathways as denoted inset: (1) the median pathway (solid lines) and interquartile ranges (IQRs, grey shadings) of all scenarios ("n" denotes the number of scenarios); (2) the median pathway of scenarios constrained by cumulative BECCS, afforestation, and DACCS limits based on expert consensus of their future CDR potential[49] ("CDR-lim."); and (3) individual AR6 illustrative mitigation pathways: "IMP-LD" – strong emphasis on energy demand reductions; "IMP-Ren" – heavy reliance on renewables; "IMP-SP" – mitigation in the context of broader sustainable development; "IMP-Neg" – extensive CDR in the energy and the industry sectors; and "IMP-GS" – less rapid and gradual strengthening of near-term mitigation actions. The projected levels of supply based on government plans and projections are shown by the black lines ("PGR21"), as estimated in the 2021 Production Gap Report[6]. See Supplementary Table 1 for the percentage reductions in coal, oil, gas supply relative to 2020 under all pathways shown.

Our findings thus highlight the tug-of-war over the fate of fossil fuels. Pulling strongly towards a rapid reduction are policies and actions that avoid and reduce emissions. Pulling the other way – or, at least, moderating the decline – is CDR and CCS, allowing for the continued use of fossil fuels, especially gas. These different approaches, while complementary and necessary from the perspective of increasing our chances of holding warming to within the Paris Agreement's temperature limits, nonetheless raises questions about which pathways may be more desirable with respect to other important societal and environmental outcomes, more precautionary with respect to safeguarding public and planetary health, and/or more feasible to attain[14,52–54].

Although researchers are only beginning to grapple with these questions, individual AR6 scenarios have been evaluated in terms of feasibility along five dimensions: geophysical, economic, technological, socio-cultural, and institutional[21]. Figure 6 shows the feasibility assessments of the pathways plotted in Fig. 5 (see Methods). We draw three main insights from this evaluation. First, all AR6 C1-C3 pathways

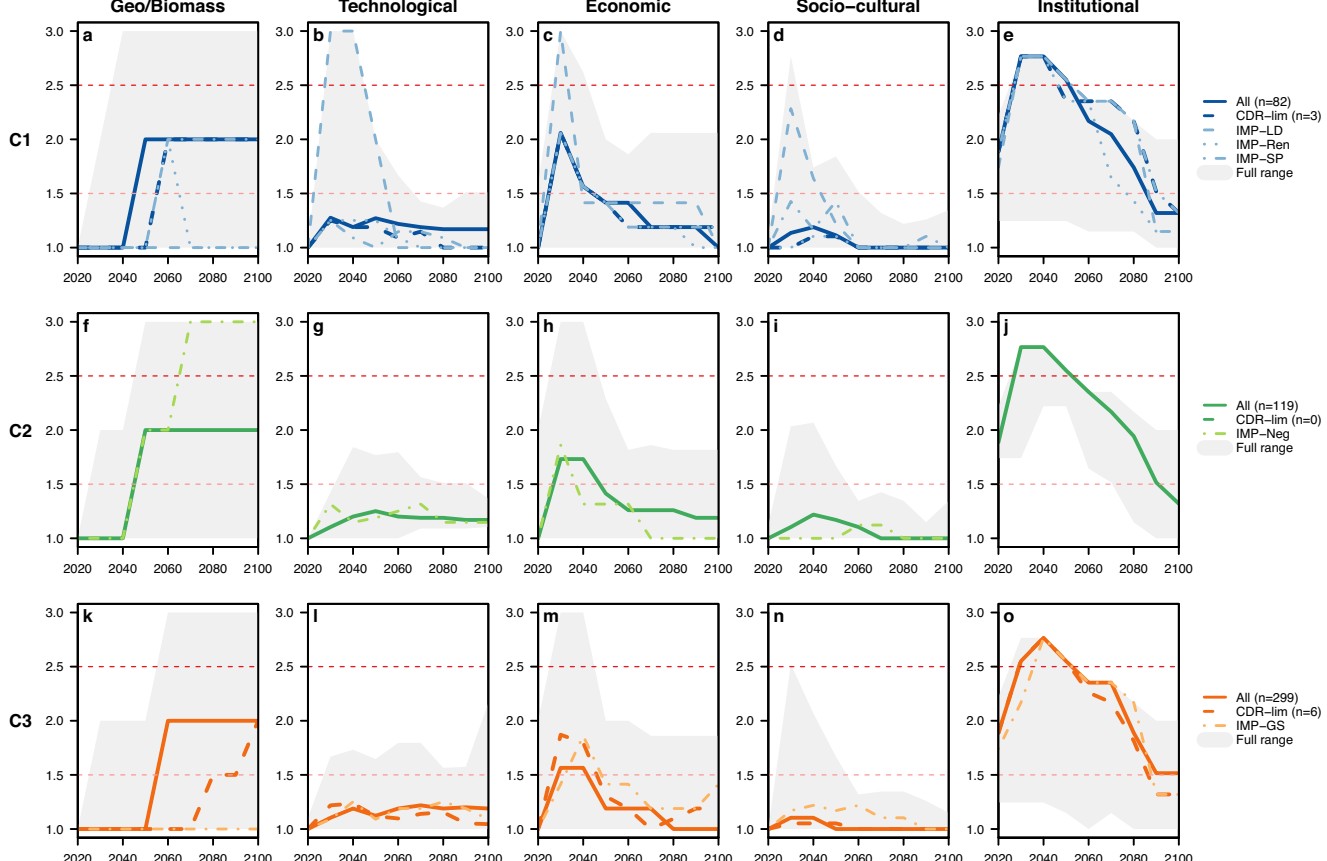

**Fig. 6 | Feasibility assessment by the IPCC AR6 WGIII of the C1–C3 pathways across five different dimensions.** Panels **a**–**o** show the 2020–2100 decadal feasibility assessment ratings by the IPCC AR6 WGIII of the C1–C3 pathways shown in Fig. 5. The grey shadings represent the full ranges of values across each of the C1–C3 ensembles. The feasibility ratings are as follows: Less than 1.5 = Low/ Plausible; >1.5–2.5 = Medium / Best case scenario; More than 2.5 = High / Unprecedented. Following Figure 3.43 of the IPCC AR6 WGIII Chapter 3, the geophysical feasibility dimension shown here is for the biomass potential indicator only. ("n" denotes the number of scenarios with available data; no data are available for assessing the institutional feasibility dimension of the C2 IMP-Neg.).

face "unprecedented" challenges in terms of governance if the continuation of current trends and existing capacities are assumed. In particular, the IMP-SP may require even greater institutional breakthroughs and coordination to ensure an equitable transition away from fossil fuels.

Second, pathways that pursue more stringent temperature outcomes are generally associated with more disruptive transformations in the near-term but then present fewer feasibility challenges and risks later. For example, among the C1 IMPs, the LD scenario, which phases out fossil fuels the fastest through substantial energy demand reductions, is associated with the highest technological, economic, and socio-cultural feasibility concerns in the near-term. In the long run, however, it presents an opportunity to avoid risks associated with delayed mitigation and reliance on uncertain CDR technologies.

Third, pathways that delay climate action and rely on extensive CDR (and consequently allow for a more gradual transition away from fossil fuels) are associated with fewer near-term technological, economic, and socio-cultural feasibility concerns, but present "unprecedented" geophysical feasibility concerns in the long-run due to how much land would be required to sustain the levels of negative emissions assumed to occur via BECCS (this is the only CDR method evaluated in the AR6 feasibility assessment). Moreover, as stated previously, only 0-15% of the C1-C3 scenarios rely on future levels of BECCS, AR, and/or DACCS below the potential cumulative limits estimated by some experts.

## Discussion

In 2021, governments reaffirmed their commitment to strive to limit global warming to 1.5 °C in the Glasgow Climate Pact[1]. Our analysis finds that there is a strong consensus in the AR6-assessed scenarios that global supply and demand of coal, oil, and gas need to decline substantially and rapidly between now and 2050 to limit warming to 1.5 °C with no or limited overshoot, and even more so if fossil-CCS and CDR technologies fail to develop at scale (i.e., by 99% for coal, 70% for oil, and 84% for gas). A recent paper estimates that, since the AR6-assessed scenarios were compiled, the remaining carbon budget for a 50% chance of limiting long-term warming to 1.5 °C has likely shrunken by 50% and now stands at 250 GtCO$_2$[55], which would necessitate even more ambitious near-term mitigation efforts across all sectors.

Many world governments seem to increasingly acknowledge the need to reduce fossil fuel demand, expand clean energy alternatives, and minimize methane emissions along the oil and gas supply chain as key climate mitigation strategies, as evidenced by the Glasgow Climate Pact's mention of moving away from (unabated) coal use and the US government's Inflation Reduction Act's focus on boosting solar and wind capacity, for example. When it comes to fossil fuel supply, however, governments' plans and projections remain vastly misaligned with pathways consistent with limiting warming to 1.5 °C or 2 °C, with estimates as of 2021[6] showing only modest decreases in coal and increasing oil and gas supply out to at least 2040 (Fig. 5, black lines), creating vast so-called "production gaps". For example, compared to the median pathway modelled by all AR6-assessed C1 scenarios, global

production levels derived from governments' energy outlooks would lead to 1200% (145 EJ) more coal, 72% (92 EJ) more oil, and 135% (110 EJ) more gas in 2040. Under the CDR-limited C1 pathways, the respective production gaps increase to 8600% (156 EJ) for coal, 81% (98 EJ) for oil, and 350% (150 EJ) for gas.

Consequently, the first policy implication of our analysis is that governments could better align their climate and energy policies by planning for and implementing an active transition away from coal, oil, and gas production along with use, starting now. Only a few fossil fuel-producing countries have begun to consider the alignment of their production and export targets with national and international climate goals, and many continue to subsidize, invest in, and plan on expanding production. This disconnect perpetuates the "production gap", undermining the Paris Agreement's long-term temperature goal and creating risks of stranded communities and assets[6]. Additionally, prior studies have shown that, to keep within the carbon budget for limiting warming to 1.5 °C with a 50% likelihood, 90% of coal and 60% of oil and gas must remain unextracted[56], while projected $CO_2$ emissions from existing fossil-fuel-production or -combustion infrastructure without additional abatement would already exceed this budget[57,58]. Combining policies to limit fossil fuel supply with policies to limit demand reduces the overall cost of achieving emission reduction goals, promotes policy coherence, helps to ensure that renewable deployment will lead to an energy transition as opposed to an addition, and directly challenges public and private vested interests who continue to lock in high fossil fuel dependence[59–64]. Furthermore, planning for and implementing a managed and equitable transition with international cooperation can help to ensure a fairer distribution of the shrinking carbon budget[65,66] and to minimize disruption for fossil fuel-dependent communities and workers[67].

Secondly, other mitigation options – such as minimizing methane emissions from fossil fuel production processes – are important but are not a substitute for directly reducing fossil fuel supply itself, which must also occur in tandem (see panel m in Fig. 4 and Supplementary Figs. 13, 14, 20–22). Some major producers – the US, Canada, Norway, Qatar, and Saudi Arabi – established the Net-Zero Producers Forum in 2021 (the UAE joined in 2022), with goals to develop mitigation strategies ranging from methane abatement to development of clean-energy and CCS technologies, but did not mention the need to reduce oil and gas production[68]. Proponents of natural gas are using the "bridge", "transition", or "cleaner" fuel narrative to justify and legitimize support and investment for expanding extraction and consumption infrastructure. However, gas could hinder or delay renewable energy transitions through locking-in fossil fuel-based technological systems and related institutions, and has been found to offer smaller benefits as an alternative to coal in terms of minimizing GHG emissions and climate impacts than originally estimated after accounting for methane leakage along the gas supply chain, which may not yet be adequately captured in existing IAMs[14]. Our analysis finds that the AR6-assessed scenarios that limit warming to 1.5 °C with no or limited overshoot and that model long-term high gas reliance are contingent upon high fossil-CCS and CDR deployment, and are most likely driven by inadequate representation of real-world constraints on $CO_2$ storage potential and on fossil fuel- and renewable-based technology innovation, adoption, diffusion, and phase-in/-out path dependencies in certain model frameworks and scenario designs. We thus conclude that our analysis lends support for the replacement of the bridge narrative, as proposed by Kemfert et al.[14], with unambiguous near- and long-term reduction benchmarks for gas production and use (e.g., by 84% between now and 2050).

Finally, the pace and extent of the required global coal, oil, and gas reduction pathways (and their relative differences) will depend on many normative factors and value-laden policy choices, some of which cannot be adequately informed by IAMs alone[69]. For example, what is the level of risk society is willing to accept in terms of the probability of limiting long-term warming to 1.5 °C? To what extent should we assume that future CDR and CCS can be developed and deployed at scale? Which climate mitigation strategy might be the most feasible and/or the most desirable? Countries' differentiated capacities and circumstances, and other equity principles, should also be considered when disaggregating global coal, oil, gas reduction benchmarks – and their relative contributions – into national pathways[70–72]. Existing low-carbon scenarios, including those analysed here, rarely incorporate equity and environmental justice considerations and are primarily driven by cost optimization[31]. For example, the rates of coal phase-out modelled by IAMs have been found to be much faster than historical precedents and would place the burden of stranded assets disproportionately on lower-income countries; greater emissions reductions in higher-income countries and faster reductions in global oil and gas could allow for a slightly slower coal phase-out in lower-income countries[73,74].

As many others have argued[26,49,54,75,76], taking a precautionary approach to minimizing climate damages would suggest opting for the most stringent temperature outcome – as reaffirmed by governments worldwide in the 2021 Glasgow Climate Pact[1] – and following pathways that limit their reliance on uncertain technologies, given that IAMs do not reflect the risk of failure of the technologies or measures on which they rely. Global rates of CCS deployment continue to fall below expectations and remain far below those modelled in the mitigation scenarios, with a total annual capacity of 45 MtCO2 as of 2021[31,77]. IAMs generally assume that $CO_2$ storage is a low-cost and globally ubiquitous resource; however, Grant et al. showed that this may lead models to substantially overestimate the role of CCS (coupled to fossil fuel and bioenergy use and/or direct air capture) – while under-utilizing renewable deployment – for decarbonization, when accounting for the technical, financial, and institutional barriers that may impose practicable limits on regional injection rates[30]. A climate mitigation strategy that entails a fossil fuel phase-out with limited CDR and CCS reliance would also bring about localized, near-term benefits from reduced air and water pollution[78,79], human rights violations[80], and biodiversity loss[81], among others. For example, exposure to outdoor fine particulate matter pollution from fossil fuel combustion is estimated to lead to around 8.7 million premature deaths worldwide each year[82], while an increasing number of studies are documenting adverse health impacts including premature birth, respiratory diseases, and cancer associated with living near fossil fuel extraction sites[83–86].

Nevertheless, aiming to limit warming to 1.5 °C with no or limited overshoot with minimal reliance on future CDR and CCS will entail some technological, economic, and socio-cultural feasibility challenges in the near-term, as this mitigation strategy implies rapid and substantial energy demand reductions, electrification and deployment of clean energy, and lifestyle shifts. But, as the IPCC notes, "feasibility concerns are context and time-dependent and malleable: enabling conditions can help overcome them"[21]. For example, rapid deployment of low-carbon technology in the near-term can help to lower the barriers for future government policy and ensure the political durability of decarbonization, as actors increasingly buy into new technologies, creating positive feedback cycles and building new low-carbon labour forces[87,88], even if actions to side-line entrenched interests and associated predatory delay will still be required[62,89]. Moreover, all scenarios that limit warming to 2 °C or below come with "unprecedented" feasibility concerns, especially in terms of institutional capacity, beyond what the IPCC could identify as plausible "best case" examples from history. Increasing international cooperation and improving the capacities of individual governments and other institutions to implement ambitious climate policies will be essential, especially for a global transition away from fossil fuels to occur in a managed and equitable manner[70,90].

While the scenarios assessed in the IPCC AR6 offer a wealth of information in terms of strategies for decarbonizing our energy

systems, it is important for decision-makers to understand their limitations and shortcomings, and the associated risks and ethical concerns of different modelled pathways, including the potential failure of CDR and CCS technologies to develop at scale[75,91], when using them to inform global and national climate policy agendas. The scenarios' differentiated alignment with other important societal and environmental outcomes like the Sustainable Development Goals (SDGs), equity, and other lines of evidence should also be evaluated[92]. Understanding the feasibility concerns of different scenarios can provide a valuable tool for identifying enabling conditions that can help to overcome them[52]. We also note that our analysis reflects only the possibilities explored in the studies underlying the AR6 scenario ensemble. Other researchers have, for example, modelled alternative degrowth scenarios in which continued growth in gross domestic production is not a prerequisite to support societal wellbeing, finding that such mitigation scenarios minimize many key feasibility and sustainability risks compared to technology-driven pathways[93].

## Methods

### The IPCC AR6 WGIII scenarios database

The raw data and metadata from all models and scenarios were downloaded from the IPCC AR6 WGIII Scenarios Database (release 1.1) hosted by the International Institute for Applied Systems Analysis[94]. Although the "IMP-Neg" scenario ("EN_NPi2020_400f_lowBECC") from the COFFEE 1.1 model is technically categorized as a C3 in the database, we categorize it as a C2 scenario here, following the IPCC AR6 WGIII Chapter 3, Table 3.2 (i.e., "The warming profile of Neg peaks around 2060 and declines to below 1.5 °C (50% likelihood) shortly after 2100. Whilst technically classified as a C3, it strongly exhibits the characteristics of C2 high overshoot scenarios"). Model outputs are available at 5- or 10-year intervals; for the latter, linear interpolation is applied to derive values at 5-year intervals for all scenarios. For all variables analysed, we estimate the cumulative 2020–2100 values from annual values derived by linear interpolation. Because the AR6 scenario ensemble is unstructured, the number of scenarios displaying a given pattern does not reflect an increased likelihood of that specific outcome. The averages and ranges of results merely reflect the options that have been explored in the studies that produced the scenarios.

Global fossil fuel supply values are taken from the "Primary Energy|Coal", "Primary Energy|Oil", and "Primary Energy|Gas" variables, which are provided in units of exajoules (EJ) per year, and reported by 97–100% of the C1–C3 scenarios (Supplementary Table 3). According to our knowledge of the COFFEE model and a 2013 survey ran by the International Institute for Applied Systems Analysis (IIASA) (V. Krey, IIASA, personal communication to R. Schaeffer, October 27, 2019), 10 of the 11 model families whose scenarios (534 out of 535 scenarios) are analysed in this paper include non-energy use under their reporting of "Primary Energy|xx": AIM, COFFEE, GCAM, GEM-E3, IMAGE, MESSAGE, POLES, REMIND, TIAM, and WITCH. (We do not know whether the EPPA model accounts for non-energy use, but there is only one C3 scenario from EPPA analysed here.) Thus, here we broadly interpret the "Primary Energy|xx" variable to represent total fossil fuel supply for all intended uses (i.e., combustion and non-combustion uses, such as for chemical or plastics feedstocks). However, the level of detail to which non-combustion uses are accounted for likely varies between different models. More consistent documentation and reporting of this issue, including by the "Final Energy|Non-Energy Use|xx" output variables (currently reported by 32–58% of the C1–C3 scenarios), would aid our analysis and interpretation. In trying to subsequently analyse which demand sectors may be heavily influencing modelled fossil fuel supply, we try to account for non-energy uses to the extent possible, using the more completely reported variable "Final Energy|Industry|xx" (where xx = Solids, Liquids, or Gases; reported by 87-93%), although this variable can include non-

fossil fuel sources. Supplementary Figs. 23 and 24 show the individual, median, and interquartile range (IQR) 2010-2100 pathways, as well as boxplot distributions of the cumulative 2020–2100 values, of the "Final Energy|Non-Energy Use|xx" and "Final Energy|Industry|xx" variables, respectively, as modelled by the C1-C3 scenarios.

### Classification and regression tree analysis (CART)

CART is a commonly used algorithm for identifying groups of scenarios by a sequence of classification rules[38]. CART can statistically demonstrate which factors are particularly important in a model or relationship in terms of explanatory power and variance while presenting the data in a way that is easily interpreted by those not well-versed in statistical analysis. We performed a CART analysis to identify the combinations of characteristics and features that are most predictive of the modelled levels of cumulative 2020-2100 supply of coal, oil, or gas in each temperature category (C1-C3 scenarios); the results are shown in Fig. 2 and Supplementary Figs. 6–11. We implemented our CART modelling using the R package *rpart*[95]. CART analysis consists of finding splits of the independent continuous or categorial variables that yield the strongest possible predictions of a dependent variable. Independent variables are not required to follow any specific distribution, and nonlinear relationships as well as interaction effects are readily captured[38].

The independent categorical and continuous variables we chose to include in our analysis reflect those that are frequently discussed in the literature[21,23,39] and/or judged to likely influence the level of fossil fuel dependence and are reported by more than two-thirds of the model-scenarios. These fall under five broad categories as follows: 1) Total primary energy supply and primary energy supply from alternative sources, including other fossil fuels; 2) End-use by different sectors (for industry use, the available variable accounts for non-energy uses but includes both fossil fuel- and bio-based sources); 3) Technologies that would enable more or less fossil fuel use, including CDR and CCS; 4) Carbon pricing – a general indicator of policy stringency and disruptiveness in the IAMs[42]; and 5) Model family and scenario project. Given a large number of individual project studies in the AR6 ensemble, the latter variable is created from the "Project_study" metadata variable by grouping projects with a relatively small number of scenarios (Supplementary Fig. 3) into one "Others" category. We use this as a proxy for capturing broad differences in scenario design but note that differences can also exist within a large project, such as ENGAGE which specifically compares net-zero budget against end-of-century budget designs[42]. For ease of interpretation and readability, we limited the tree growth by pruning the tree using *k*-fold cross-validation and limiting the minimum number of scenarios in a leaf node to 5% of the total in each category. Following general convention, we used a value of $k = 10$ and selected the simplest tree within one standard error of the best tree (lowest cross-validated relative error)[96].

### Analysis of variance (ANOVA) and K-means clustering

To identify the factors that contribute most to the variability in modelled gas supply between 2020 and 2100 in each of the C1-C3 scenario ensemble, we performed an analysis of variance (ANOVA), following Guivarch and Monjon (2017)[97]. The contributions of each driver over time is based on the partitioning of the sum of squares in a linear regression. The total sum of squares (i.e. the variance of the results, multiplied by the number of scenarios minus one) can be partitioned into the explained sum of squares linked to each uncertainty driver and the residual sum of squares. This partitioning of the sum of squares is computed at each decadal interval for the ensemble of scenarios. It gives the fraction of variance explained by each uncertainty driver at each time step. To minimize multi-collinearity between the independent variables, we include only eight categorical and continuous independent variables identified as most important in our CART analysis: "Model family", "Scenario project", "Carbon Sequestration|CCS|

Biomass","Carbon Sequestration|CCS|Fossil", "Emissions|CO2 | AFOLU", "Final Energy|Industry|Gases*" (which is highly correlated with "Final Energy|Transportation|Gases*"), "Price|Carbon", and "Primary Energy|Non-Biomass Renewables". The Variance Inflation Factor in our ANOVA models for all fuels, categories, and years did not exceed 3.1. (*These final energy variables include both fossil fuel- and bio-based sources.)

We use the R package *cluster* to group the scenarios in each of the C1–C3 categories into three different types based on their modelled values of gas supply in 2030, 2050, 2075, and 2100 by *k*-means clustering[98]. This is one of the most commonly used unsupervised machine learning algorithm for partitioning a given data set into groups, where *k* represents the number of groups pre-specified. It classifies objects in multiple groups (i.e., clusters), such that objects within the same cluster are as similar as possible, whereas objects from different clusters are as dissimilar as possible. Based on Fig. 1, we observe three general patterns of gas supply trajectories in each of the C1-C3 categories and therefore chose a value of $k = 3$.

### Sensitivity analysis to uncertainties in CDR potential
In Fig. 3, we limited the CDR levels that a scenario can rely on based on the median cumulative 2020-2100 feasible potential derived from an expert consensus survey conducted by Grant et al.[49] for three CDR methods: no more than 196, 224, and 320 $GtCO_2$ of BECCS, afforestation, and DACCS, respectively. This feasible potential reflects a combination of technical constraints (e.g., resource availability in terms of how much biomass can be grown for energy[99]), sustainability considerations (e.g., intensive land use for bioenergy or $CO_2$ sequestration by the AFOLU sector could lead to land degradation, food insecurity, biodiversity loss, and water scarcity[28,29]), as well as social and governance concerns[49].

We use the values reported under the "Carbon Sequestration|CCS|Biomass" variable (reported by 93–98% across the C1-C3 scenarios) to constrain BECCS and "Carbon Sequestration|Direct Air Capture" to constrain DACCS (reported by 25–41%). Given that 62–66% of the scenarios report "Carbon Sequestration|Land Use" and 39–55% report "Carbon Sequestration|Land Use|Afforestation", we use the former variable representing total sequestration by all land use methods when the latter is not available to increase the coverage for constraining afforestation. The resulting median pathways of coal, oil, and gas supply are not affected by applying this proxy method compared to using the "Carbon Sequestration|Land Use|Afforestation" variable alone (see Supplementary Fig. 25). If a model-scenario reports zero for a particular variable, we assume that CDR method is modelled and the output is zero. If the variable is not reported, we exclude that model-scenario from the CDR-limited subset given that CDR may still be used at levels exceeding the thresholds. For example, all model-scenarios report net AFOLU emissions, but only 39–55% report afforestation under the "Carbon Sequestration|Land Use|Afforestation" variable, meaning that some models implicitly rely on afforestation but do not report it. Given this lack of reporting consistency, especially for land use sequestration, some errors may have been introduced in our interpretation.

The median values (and interquartile ranges) of cumulative 2020-2100 CDR via BECCS, afforestation, and DACCS in the C1-C3 scenarios are 328 (253-527) $GtCO_2$, 217 (136-240) $GtCO_2$, and 29 (0-192) $GtCO_2$, respectively (see Table 3.5 of the AR6 WGIII Chapter 3 for details[21]). We find that 4 out of 26 C1, 0 out of 16 C2, and 6 out of 62 C3 scenarios that report the relevant variables do not exceed the limits based on Grant et al.[49].

### Feasibility assessment of the mitigation scenarios
The IPCC AR6 WGIII assessed the feasibility of mitigation scenarios along five dimensions: geophysical, technological, economic,

institutional, and socio-cultural, following the methods developed by Brutschin et al.[52]. Table II.1 of the IPCC AR6 WGIII Annex III summarizes the indicators used for each dimension, and their respective thresholds for the categorization of values exceeding medium or high levels of concern[100]. For a given indicator, a rating of 1-3 is assigned as follows: below the medium threshold value = 1; between the medium and high threshold values = 2; and above the high threshold value = 3. For a given feasibility dimension, the relevant indicator ratings are then aggregated using the geometric mean, with the resulting mean categorized as follows: <1.5 = "plausible"; 1.5-2.5 = "best case"; and >2.5 = "unprecedented".

### Data limitations
In general, our analyses and scenario interpretations are constrained by data availability: not all variables are consistently reported across the AR6 model-scenarios (Supplementary Table 3). Consequently, there could be a model sample bias in the median and distributions shown for a given variable, or in the independent variables identified as the most important predictors in the CART and ANOVA analyses. Increased transparency and reporting of model inputs, parameters, assumptions, and outputs, such as on the costs of different energy technologies, carbon prices, the levels and specific methods of CDR assumed, and differentiated fossil fuel end-uses by sector, across all IAMs would increase their interpretability for policy analysis.

## Data availability
All raw data and metadata of the scenarios analysed in this study were downloaded from the IPCC AR6 Scenarios Database (release 1.1) hosted by the International Institute for Applied Systems Analysis, available at: https://data.ece.iiasa.ac.at/ar6/. Data on the feasibility assessments of the C1-C3 scenarios used in this study, and all data generated in this study, can be downloaded from https://github.com/ploy-a/NCOMMS-AR6-FF.

## Code availability
Scripts to generate the analyses and figures in the manuscript are available from https://github.com/ploy-a/NCOMMS-AR6-FF.

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

## Acknowledgements

P.A. thanks Michael Lazarus (Stockholm Environment Institute) for helpful feedback on manuscript drafts. P.A. and P.E. acknowledge funding support from the Energy Transition Fund (Grant number G-21-2122456), a project of Rockefeller Philanthropy Advisors (RPA). R.S. acknowledges funding support from Conselho Nacional de Desenvolvimento Científico e Tecnológico (CNPq), Brazil (Grant number 310992/2020-6). C.G. acknowledges funding support from the European Union's Horizon Europe research and innovation programme under grant agreement No. 101081604 (PRISMA). Any errors are the sole responsibility of the authors.

## Author contributions

P.A. led the analysis and manuscript preparation with help from P.E., P.A., P.E., C.G., R.S., and S.P. designed the paper concept. E.B. generated and provided data on the feasibility assessment of the IPCC AR6 scenarios database. All authors contributed to the text.

## Competing interests

The authors declare no competing interests.
