## [Peer Review File · Nature Communications]

Global fossil fuel reduction pathways under different climate mitigation strategies and ambitionsREVIEWER COMMENTS

Reviewer #1 (Remarks to the Author):

Overall comments

This paper, which undertakes a detailed analysis of the global fossil fuel use implications of the IPCC AR6 database mitigation scenarios in the C1 to C3 categories (i.e. 1.5-2C), is a very clear and enjoyable read. It is highly policy-relevant, and uses a number of nice analyses to demonstrate the detailed dynamics of the pathways analysed.

I felt the Abstract, having read the whole paper, didn't do the paper justice, and I was quite sceptical about whether the paper was sufficiently high quality to make it into Nature Communications based on this. So, however possible within the word limit, I would recommend the Abstract highlight how, aside from just drawing statistics from the AR6 database, they have been subjected to a regression tree analysis to see what drives them, a filtering based on CDR limits and comparison to planned fossil fuel production levels, and a feasibility analysis. In other words, something that highlights that this isn't just a paper made out of the kind of very nice R / Python chart-based Tweets that some widely-followed academics regularly publish to show interesting insights from IPCC scenario databases!

I think overall the paper has (just) won me over that there's enough novelty and quality of analysis to potentially make it into this journal. I made several comments below, some of which might not be trivial to deal with. But if they are, then I think this will end up being a valuable piece of work that helps highlight the challenge of shifting away from fossil fuels.

Having said that, the messages are all kind of obvious, at this stage of the game, and the methods used (analysis of the IPCC database, filtering according to various criteria, comparison to production levels, feasibility assessment) aren't really novel. It's just the overall combination of them, combined with the quality of the analysis and writing, that arguably gets this over the line for me. Sorry to be so equivocal – I hope you can see why.

Specific comments

Line 33-34 – Please spell out a bit more explicitly what the consequence of the Russia invasion is / could be for fossil fuels. There are conflicting views about whether this will serve to accelerate the clean energy transition or lock in LNG production and revive coal use in various parts of the world, so just saying this (ideally with some references) would be better than the slightly tantalising statement here.

Line 40 – There are 1,686 temperature-vetted scenarios, I believe. What does the 1,202 refer to? Is that what Ref 8 says? In any case, it would be good to say explicitly what these 1,202 from the larger number of scenarios in the database refers to.

Line 80 – I would contest that the AR6 database allows a communication of the range of possibilities, since the scenarios and pathways in the database don't cover all potential possible / plausible scenarios.

Line 93 – I got a bit confused with the two dotted lines on the oil and gas charts in Figure 1. Can they be coloured so as to correspond to the relevant scenario category (i.e. C1, 2, 3)? I assume there are supposed to be three dotted lines per chart, showing the peak year for each scenario category? Maybe (given that they seem to overlap heavily) they could be shown in different way.

Line 103-124 – I didn't see anything surprising or counter-intuitive in these results. I wondered whether the authors should therefore comment on this (i.e. when discussing the scenarios with higher coal, oil, gas supply). Also, good to just make a bit clearer here that the insights are derived from the CART analysis (I know the relevant Supplementary figures are referenced, but even so, this would

help make clearer in the main manuscript what the point of doing the CART analysis was).

Line 123 – I would have liked to see a reference to these highly contested policy debates around gas.

Line 132 – What's a decent justification for 2030 as the comparison year to 2100? And why 2100?

Line 136 – Figure 2 is nice, but a key thing I was wondering was what the high and low gas scenarios might show in terms of primary gas energy with and without CCS. The reason I raise this is because it's tempting to think that high gas scenarios are that way because they just have a big role for CCS, whereas low gas scenarios don't. Figure S2 shows gas w/ and w/out CCS, so why not show that here? I think it would be good to.

Line 175-178 – This is a good point, and perhaps gets to the heart of why this analysis is very interesting but maybe not as novel (or maybe even useful) in its findings as it could be: it shows the outputs of IAMs, indicating (fairly intuitive) trends in how higher and lower gas (and indeed other fossil fuel) levels relate to other "levers" of decarbonisation. But it doesn't have the ability to explain what drives the different results: particular assumptions from modellers, real-world constraints in roll-out rates of low-carbon technologies, some underlying assumptions which show (for example in the low-gas scenarios) that once gas is gone, then path-dependencies mean it won't come back, versus no such assumptions in the high-gas scenarios? On the latter point, I'd be interested to see a comment from the authors on whether they think it's realistic to see an (unabated, in many cases like transport) gas rebound later in the century, given the climate emergency we're in, and the pressure to shift away from fossil fuel industries.

Line 266 – Figure 4 is very nice, but made me realise that there is scope to do more with this paper, in terms of highlighting particular regions in which the feasibility challenges really become difficult. Could the authors, at least for the institutional indicator which I understand is in any case aggregated at the global level from underlying country level data, analyse where the particular institutional constraints / challenges are coming? This would help highlight where additional international funding / support / capacity building should be aimed. I think there's only really one comment on national pathways, around line 308, and more could be done here, even within the tight constraints of the 5-region IPCC database.

Line 266 – Is the shaded area in Figure 4 the interquartile range?

Reviewer #2 (Remarks to the Author):

This is an article that needs to be written (and should have been written years ago), but I feel this article needs a more solid analytical and statistical foundation. The article is primarily based on summary statistics of the AR6 scenarios database, with a little additional analysis, but I think a more robust statistical analysis is needed (for Nature Communications at least). There are two main issues associated with this: 1) It is stated that the AR6 scenarios database is unstructured, but nothing is done about it, and 2) There is a CART analysis, but it is hard to see how it informs the analysis. I have lots of comments below, but these are mainly suggestions for improvement and should be seen in the context of making this a stronger and more robust article. The comments are in order they appear in the article, not ranked in order of importance.

Longer comments:

1. The article says nothing about equity, but it would be expected that including equity considerations will change the global pathways. Since coal is now primarily used in non-OECD, greater equity will lead to slower reductions in coal and greater reductions in oil and gas (which is primarily in OECD).

You can see this in some of the figures of Bauer et al 2020, mainly SI, <https://www.nature.com/articles/s41586-020-2982-5>. It is also rather intuitive. You could try and talk your way around this one, but it is sort of a fundamental issue. The analysis is based on the AR6 database, which includes almost no scenarios with equity considerations. Not only is the database unstructured, but it also misses some substantial and fundamentally important considerations. You need to be explicit about these shortcomings, and perhaps draw on relevant literature to make broader points.

2. The statement on lines 39 and 40 is factually correct, but misleading. As reference 8, Figures 3.1 and 3.2 show, the scenarios database is extremely skewed to some models and projects. This skewness is even worse if one looks individually at C1, C2, and C3 separately. As Guivarch et al 2022 suggests, "several important aspects have to be addressed" when dealing with such a database, and I don't see that you have addressed these. Ok, AR6 Chapter 3 did not address these, and got away with it, but as a scientific community, we must do better. I think you need to have a solid discussion on how the bias in the database may affect your results. When you look at C1, are you showing me anymore than REMIND (41 of 97 scenarios). Or C3, 176 of 311 are from ENGAGE, with its very particular scenario design. To be clear, 1202 may be the number of scenarios and 44 may be the number of models, but that tells me nothing about the underlying statistical distribution, which largely renders summary statistics from the database unusable. Your challenge is to prove me wrong!

3. Lines 52-54 are a little misleading, as you write in lines 400+. Chapter 3 Table 3.5 does not show total CDR, it was removed from the final version (see corrigenda), because the IAMers did not know what AFOLU or Afforestation represented in different models. In any case, your text gives the impression that DACCS is more prevalent than it is. You could perhaps bring some text from the methods forward here to be clear about this. As you write in the methods, all models have AFOLU, but roughly half afforestation, so really, models all implicitly include afforestation, we just don't know how much. All models, basically, include BECCS. DACCS is not so well represented. As for the previous comments, these numbers are biased by models too. If you remove one model from the database, maybe the statistics are rather different! This is problematic when you talk about the limited CDR pathways towards the end, where you are looking at a far more skewed version of the already very skewed database!

4. Lines 71+. Since this is not an IPCC report, I feel my duty to ask for justification of using median and interquartile range (IQR). What is your statistical justification? What about 0-100 (range), 5-95, 33-66, 25-75, median, mean, etc. Is there something about the statistical distribution which leads you to use IQR? You are not plotting half the data! Page 3 is all discussing the median. But what does the median even mean in this database? Why even plot Figure 1 in that style? You talk about % reductions in particular years, so maybe a more statistically prudent way of doing this is to show a histogram of time slices, which would somewhat avoid issues on choosing IQR or something else? What is the histogram of coal in 2050 (for example)? You also reference Guivarch et al, saying that the approach you have taken "has limitations", but have you tried to address those limitations? We now know the database is unstructured and biased, so I don't think it is possible any longer to just brush that aside! You have to draw to address those limitations as best you can!

5. Lines 199-210 you write "...higher gas supply also appear to rely on greater CDR...". I have problems with "appear". You should be able to show this? Perhaps this is a lead into the next section, but the next section does not seem to draw on the CART. CART is not mentioned until the methods, and I don't really see how the analysis from lines 125+ is informed by the CART. Are the lines 126-130 the CART? That is not obvious. The split of gas into two parts is also a little arbitrary. All choices will be arbitrary, but since you use cumulative energy use to 2100, you could split the 2100 distribution 50-50. Or did you make the 2030 choice because that shows something special? Again, on Figure 2, not sure how helpful the IQR is, but I understand you have to make a choice. Basically, I am lacking the statistical justification for the choices you are taking.

6. Figure 3. I am going to have a lot to say about this figure... You may see bigger biases here, and model and project fingerprints. A key figure that is missing is total primary or final energy. I see you have components of demand, but you could probably drop those and just have PE or FE or both. That, I think, will explain quite a lot of your first order variation? It may be the top layer of your CART, if you restrict the variables? There are a few obvious fingerprints in the figures. The gas going down then up, that is MESSAGE. The high nuclear, that is MESSAGE and to a lesser degree GCAM. So high gas implies high nuclear? Why? Is there a causal reason for that, or is it just because MESSAGE is dominating that part of the distribution? Try taking MESSAGE out and see what the figures look like. Repeat for all models. Did you try total CCS (bio + fossil), why is fossil CCS so important but not BECCS? Transport demand gases, that is MESSAGE! You are likely to also find a quite strong ENGAGE imprint on this figure, as the experimental protocol of ENGAGE will cause the bifurcation you have in panel a. Basically, my concern here, is that you are just showing the outcomes of MESSAGE or ENGAGE. In all the figures I plot, nearly all the purple outliers show MESSAGE! That may have value, just like selecting an IMP might have value, but it might cloud your policy recommendations. If you select another variable or another Cx, then you will probably pick up a fingerprint of REMIND as it is so dominant, etc. The carbon price fingerprint, that may well be heavily affected by the ENGAGE model protocol, particularly in C2 and C3 where ENGAGE is more dominant. Etc. I think some effort is needed in Figure 3 to deal with model and project bias, as this is the only way to get to some of the most important policy conclusions, I think.

7. I am a little uneasy on the IMP section. The IMPs are drawing out characteristics of one scenario, but it is unclear if that one scenario is more representative of the desired characteristics or the model or the intercomparison. For example, do all low demand scenarios have the same characteristics as IMP-LD, or is this heavily influenced by MESSAGE. There also seems to be a slight contradiction, in that you start off by analysing the database to get a spread across scenarios, but then fall back on to individual IMPs. You could go back another step, and do more stylised "if gas use is higher, then it needs more CCS, more CDR, or other fossil fuels to go down faster for a fixed carbon budget". These are just random thoughts on the IMPs, as to me it was not clear how they fit the overall narrative. In my view I think it is worth reflecting whether there is value in discussing the IMPs or expanding other parts of the analysis.

8. Line 369: "model itself is also an important predictor". I think you need more justification on this point. If the model is an important predictor, how can you ignore it. There might be grounds, but you need to convince me of that.

Minor comments:

9. Lines 33 and 34, references 4 and 5. Particularly reference 4, you would be much better off to use Friedlingstein et al 2022, Global Carbon Budget 2022, as it is a) peer reviewed, b) more recent than the IEA analysis. You could also use this for the statement on Ukraine, but the WEO may fit your needs better?

10. Line 53, reference 11. Fine reference, but it is not based on AR6 which is your focus. Is there a relevant reference of AR6 showing CDR is not much different to SR15? Maybe just also add a ref to Ch3?

11. Lines 75. There is at least one study that would say many C1, C2, and C3 are not Paris compliant. <https://www.nature.com/articles/s43247-022-00467-w>

12. Figure 1: I guess the vertical dotted lines sit on top of each other. Each figure should have three? Perhaps state that?

13. Lines 174+. But, you should be able to assess this?

14. Figure 4. If I understand correctly, you took all the criteria from Chapter 3, and implemented it?

15. Lines 275. Well, that is sort of a sweeping statement. Didn't your figures, particularly if you look at the range (and not IQR), show that gas does not need to decline substantially and rapidly? In my view, coal, oil and gas do need to follow what is prescribed in your first sentence, but I do not think your analysis supports that (particularly for gas).

16. Line 282. Sure, fine, and likely true, but did your analysis show this? I don't think you analysed methane? (hint, what happens if you put methane in your CART...)

17. Line 290. 8600%. Sort of meaningless. This is small divided by small, right?

18. Line 303. "wind down", I am fine with it, but considering this is an article about energy, I suspect many like me will read "wind (power)". Perhaps use "reduce"...

19. Overall, I think you could shorten the discussion. Not that you say anything wrong per se, but many statements are not necessarily supported by your analysis.

20. Line 348, footnote 1. Quite an important point. Most of the oil in 2050 and beyond is likely for feedstocks. Though, I suspect you can not find out the reporting quality. There is one non energy variable, perhaps in final energy, but I am not sure how many models report it.

21. Lines 368. "Carbon pricing", within each category, like C1, C2, C3, the carbon pricing might be very heavily affected by the ENGAGE scenario protocol.

22. Line 365: CART. I did not really see how this was used to inform the analysis. There are a bunch of figures, but how did you use this knowledge in your analysis?

23. Lines 405. C1, 4 out of 26 out of 97. So only 26 scenarios had sufficient data, and this is perhaps highly skew by model, and therefore highly biased. Likewise for C2 and C3. At least show how many scenarios did not have the data!

Supp Info

24. Line 8: AR6 assessment, not really the database?

25. IMP-LD, Line 13. This scenario has high carbon price. Does that mean all LD scenarios have high carbon price? Or is this something about this scenario in MESSAGE? For the IMPs to be useful in this way, you need to be confident they are representative. I do not see that the Ch3 approach of selecting IMPs was so systematic, such as taking a CART and letting the IMPs fall out of the CART. I would be more convinced of the IMPs, if you showed that they were consistent with the CART, for example, or even if you had your own IMPs come out of your analysis.

26. Figure S2. Mentions IQR in caption, but I don't see it in the figure.

27. I am not an expert on CARTs, but I guess in context, you have to tell the system what variables to look at. You have used a mix of variables, some of which might be reported. Some will overlap, things like SE and PE, BE and BECCS, etc. To what degree is it important to decide what variables that CART can consider? But most importantly, what did I learn from the CART, and how is it used in the main paper. This I missed!

Reviewer #3 (Remarks to the Author):

Many thanks to the authors for this paper. It provides some interesting insights into the prospects for fossil fuel demand under a range of different climate scenarios.

There are four main analytical elements to the paper as I see it: a high-level analysis of the IPCC scenario, a grouping of some of these scenarios according to gas use, a Classification and regression tree (CART) analysis, and feasibility assessments of the pathways.

The high-level analysis of the IPCC scenario database looks robust and I have not seen this published elsewhere. I was not able to follow fully the Classification and regression tree analysis (CART) and whether this has been carried out correctly as this is outside my area of expertise, but the results make intuitive sense. I did not understand the feasibility assessments of the pathways (underpinning Figure 4), as these are not really described in any detail in the paper or the supplementary information. Much more description of what this is based on and where the results have come from is needed.

I am not convinced by the analysis on gas. In general, this section requires a much more balanced discussion on what are the results and a better justification for the conclusions. The authors conclude (in the abstract) that "gas remains a significant energy source only if its use can be coupled to carbon capture and storage, combined with relatively limited renewables deployment and extensive land-based carbon sequestration". It is not clear how these specific elements were selected: Figure 2 shows there are elements that appear to have a much larger impact on the role of gas. For example, differences in nuclear and bioenergy use are much larger between the two scenario groups than renewables. Scenarios with more gas, have much lower long-term CO₂ price and these elements seem much more noteworthy and significant than the ones highlighted by the authors.

The conclusion on land-based carbon sequestration is especially weak as the figure shows that while they have more land-based carbon sequestration, they have much lower BECCS. The sum of BECCS and AFLOU (i.e. total carbon dioxide removal) does not look materially different between the two scenario groupings.

Given what is shown, there is a strong possibility that the differences between the gas groups are more a function of the IAMs that are contributing more or fewer scenarios to each scenario group rather than real differences in findings. For example, it may be that one model contributes the majority of scenarios in one group and this model has an inherent inclination to have higher energy demand, less gas and lower CO₂ prices.

A second area that needs attention is the implications of their findings on the non-energy use of fossil fuels (e.g. the use of oil as a petrochemical feedstock). This is mentioned briefly in the methods section, but given the importance of the sector to fossil fuel demand, it needs to be analysed in much more detail.

Finally, the authors conclude that "governments need to urgently plan for and implement a global wind-down of coal, oil, and gas, starting now." It is not clear how this conclusion is reached: it may well be the case that a wind down in fossil fuel use can be achieved by boosting clean energy technology deployment and fossil fuel supply falls as a consequence. While a plan for reducing fossil fuel use would be helpful to avoid negative impacts, the paper does not justify why implementing a wind down is necessary.

In general, this could be an interesting paper, but I am worried that some of the findings and the conclusions suffer from selection bias.

RESPONSE TO REVIEWER COMMENTS

We would like to thank all three referees for their careful review and for their constructive and thoughtful comments. Below, we respond to the comments in detail. The referees' comments are in plain text, our responses are in blue text, and changes to the manuscript are *in italics*. In our responses we refer to page and line numbers according to our revised manuscript ("03_AR6-FF_revised_tracked.pdf").

Please note that we have also revised our analyses and all figures using the latest, updated version 1.1 of the AR6 scenarios database.

#####

Reviewer #1 (Remarks to the Author):

Overall comments

This paper, which undertakes a detailed analysis of the global fossil fuel use implications of the IPCC AR6 database mitigation scenarios in the C1 to C3 categories (i.e. 1.5-2C), is a very clear and enjoyable read. It is highly policy-relevant, and uses a number of nice analyses to demonstrate the detailed dynamics of the pathways analysed.

I felt the Abstract, having read the whole paper, didn't do the paper justice, and I was quite sceptical about whether the paper was sufficiently high quality to make it into Nature Communications based on this. So, however possible within the word limit, I would recommend the Abstract highlight how, aside from just drawing statistics from the AR6 database, they have been subjected to a regression tree analysis to see what drives them, a filtering based on CDR limits and comparison to planned fossil fuel production levels, and a feasibility analysis. In other words, something that highlights that this isn't just a paper made out of the kind of very nice R / Python chart-based Tweets that some widely-followed academics regularly publish to show interesting insights from IPCC scenario databases!

I think overall the paper has (just) won me over that there's enough novelty and quality of analysis to potentially make it into this journal. I made several comments below, some of which might not be trivial to deal with. But if they are, then I think this will end up being a valuable piece of work that helps highlight the challenge of shifting away from fossil fuels.

Having said that, the messages are all kind of obvious, at this stage of the game, and the methods used (analysis of the IPCC database, filtering according to various criteria, comparison to production levels, feasibility assessment) aren't really novel. It's just the overall combination of them, combined with the quality of the analysis and writing, that arguably gets this over the line for me. Sorry to be so equivocal – I hope you can see why.

Thank you very much for your balanced review and helpful comments and suggestions. We have edited the abstract following your recommendations, and addressed your specific comments one-by-one below.

Specific comments

Line 33-34 – Please spell out a bit more explicitly what the consequence of the Russia invasion is / could be for fossil fuels. There are conflicting views about whether this will serve to accelerate the clean energy transition or lock in LNG production and revive coal use in various parts of the world, so just saying this (ideally with some references) would be better than the slightly tantalising statement here.

Thank you for your helpful suggestion. We have edited the text accordingly:

Page 1, line 37: “At the same time, global fossil fuel-derived carbon dioxide (CO₂) emissions rose to the highest level in history in 2021-2022⁵, and the ongoing global energy crisis has sparked a reappraisal of energy policies and priorities worldwide. It remains unclear whether this will serve to accelerate the clean energy transition or lock in fossil fuel dependence⁶, with many countries expanding fossil gas production or import capacity and others reviving coal use in their short-term responses⁷. This occurs against a backdrop of long-standing and growing tensions as to whether fossil gas is a “bridge fuel” to a low-carbon future or a “bridge to nowhere”⁸⁻¹².

Line 40 – There are 1,686 temperature-vetted scenarios, I believe. What does the 1,202 refer to? Is that what Ref 8 says? In any case, it would be good to say explicitly what these 1,202 from the larger number of scenarios in the database refers to.

We have edited the text to be more accurate (please see below). Based on Chapter 3, page 3-116: “The AR6 scenario database contains 3,131 scenarios of which 2,425 with global scope were considered by this chapter, generated by almost 100 different model versions, from more than 50 model families. Of the 1,686 vetted scenarios, 1,202 provided sufficient information for a climate categorization.”

Page 2, line 23: “The Working Group III (WGIII) contribution to the Intergovernmental Panel on Climate Change (IPCC)’s Sixth Assessment Report (AR6) compiled 3,131 scenarios generated by almost 100 different model versions from more than 50 model families, with varying regional scope and temperature outcomes¹⁹”

Page 2, line 41 – Page 3, line 8: “Chapter 3 of the IPCC AR6 WGIII report vetted 1,686 global scenarios, of which 1,202 provided sufficient information for a temperature outcome categorization, ranging from “C1” to “C8”¹⁹. We interpret the C4-C8 scenarios to be inconsistent with the Paris Agreement’s long-term temperature goal and focus on the C1-C3 categories with the lowest temperature outcomes: “C1” – limit warming to 1.5°C in 2100 with a likelihood greater than 50%, with no or limited overshoot; “C2” – limit warming to 1.5°C in 2100 with a likelihood greater than 50%, with high overshoot; and “C3” – limit peak warming to 2°C with a likelihood greater than 67%¹⁹. Here we analyze all scenarios in the C1-C3 ensemble and consider them to be relevant to the Paris Agreement’s temperature limits, even if some individual scenarios may be considered to not be fully Paris-compliant depending on one’s interpretation of its long-term temperature goal and of its other goals, as well as judgement on the probability, overshoot, and timing of the temperature change³². Of the 541 vetted C1-C3 scenarios, 94 C1, 131 C2, and 310 C3 scenarios report “primary energy supply” from coal, oil, and gas. Since these variables can include non-energy uses for fossil fuels, we interpret them as total supply for all uses (see Methods).”

Line 80 – I would contest that the AR6 database allows a communication of the range of possibilities, since the scenarios and pathways in the database don’t cover all potential possible / plausible scenarios.

We have edited this text for clarity. What we meant here was that our approach to focus on the medians and IQRs of the full AR6 ensemble in our main figures has its limitations but nonetheless

provides one way of communicating the range of results seen across the scenarios within each climate category (i.e., it's the range of possibilities within AR6 ensemble, we fully agree that this doesn't capture all possibilities).

Page 3, line 10: Figure 1 shows the individual, median, and interquartile range (IQR) 2010-2100 pathways, as well as boxplot distributions of the cumulative 2020-2100 values, of the global coal, oil, and gas supply as modelled by the C1-C3 scenarios (see Figures S1-S2 for boxplot distributions of the annual values at certain years and of the peak years of supply, respectively). Showing the median pathway provides one way to succinctly communicate the average trajectory within a given scenario ensemble. However, this approach has limitations³³, especially because the AR6 ensemble is unstructured and does not represent a statistical sample: it merely reflects the options that have been explored in the underlying studies, which are represented by different model families and scenario designs and protocols to varying degrees (Figure S3). Given the diversity in the model outputs, the median pathways shown in Figure 1 should be considered for illustrative purposes only, as we will later explore how subsets of or individual scenarios diverge from them."

We have also added this discussion point in the Discussion section:

Page 16, line 3: "We also note that our analysis reflects only the possibilities explored in the studies underlying the AR6 scenarios ensemble. Other researchers have, for example, modelled alternative "degrowth" scenarios in which continued growth in gross domestic production is not a prerequisite to support societal wellbeing, finding that such scenarios minimize many key feasibility and sustainability risks compared to technology-driven pathways⁷⁹."

Line 93 – I got a bit confused with the two dotted lines on the oil and gas charts in Figure 1. Can they be coloured so as to correspond to the relevant scenario category (i.e. C1, 2, 3)? I assume there are supposed to be three dotted lines per chart, showing the peak year for each scenario category? Maybe (given that they seem to overlap heavily) they could be shown in different way.

We think that colouring these dotted lines could add even more confusion to this figure, and so we have decided to remove them and to also separate out the C1-C3 pathways into separate subplots for clarity in a revised version of Figure 1. The peak years of different pathways are still listed in Supplemental Table S1.

Line 103-124 – I didn't see anything surprising or counter-intuitive in these results. I wondered whether the authors should therefore comment on this (i.e. when discussing the scenarios with higher coal, oil, gas supply). Also, good to just make a bit clearer here that the insights are derived from the CART analysis (I know the relevant Supplementary figures are referenced, but even so, this would help make clearer in the main manuscript what the point of doing the CART analysis was).

While the results may not be surprising, our primary aim with the CART analysis (and this paper) is to provide a useful and accessible explanation to a broad and interdisciplinary audience of researchers beyond the academic community directly engaged in developing or using the AR6 scenarios, as well as policymakers and non-academics, as to what these scenarios say about fossil fuel reduction pathways consistent with limiting warming below 2°C and to explain the diversity seen in the modelled outputs, especially as to why some scenarios see gas phasing out whilst others see gas remaining a major part of the energy mix. Nevertheless, we agree that the purpose of and insights drawn from our CART analysis

need to be made clearer. We have revised the text quite substantially in this section, and moved the CART figures for C1 coal, oil, and gas from the Supplement into the main paper as Figure 2.

Line 123 – I would have liked to see a reference to these highly contested policy debates around gas.

We have moved this text to earlier in the manuscript and added the following text and references:

Page 1, line 39: “...It remains unclear whether this will serve to accelerate the clean energy transition or lock in fossil fuel dependence⁶, with many countries expanding fossil gas production or import capacity and others reviving coal use in their short-term responses⁷. This occurs against a backdrop of long-standing and growing tensions as to whether fossil gas is a “bridge fuel” to a low-carbon future or a “bridge to nowhere”^{8–12}.

7. IEA. *World Energy Outlook 2022*. <https://www.iea.org/reports/world-energy-outlook-2022> (2022).
8. Janzwood, A. & Millar, H. *Bridge fuel feuds: The competing interpretive politics of natural gas in Canada*. *Energy Research & Social Science* **88**, 102526 (2022).
9. Landrigan, P. J., Frumkin, H. & Lundberg, B. E. *The False Promise of Natural Gas*. *New England Journal of Medicine* **382**, 104–107 (2020).
10. Brauers, H. *Natural gas as a barrier to sustainability transitions? A systematic mapping of the risks and challenges*. *Energy Research & Social Science* **89**, 102538 (2022).
11. McGlade, C., Pye, S., Ekins, P., Bradshaw, M. & Watson, J. *The future role of natural gas in the UK: a bridge to nowhere?* *Energy Policy* **113**, 454–465 (2018).
12. GECF. *Long-Term Strategy of the Gas Exporting Countries Forum (Second Edition)*. <https://www.gecf.org/about/long-term-strategy.aspx> (2022).

Line 132 – What’s a decent justification for 2030 as the comparison year to 2100? And why 2100?

We have revised this analysis to be more statistically robust and applied k-means clustering to group the gas pathways into three different typologies based on their modelled values in 2030, 2050, 2075, and 2100. Please see our revised manuscript and methods for details.

Line 136 – Figure 2 is nice, but a key thing I was wondering was what the high and low gas scenarios might show in terms of primary gas energy with and without CCS. The reason I raise this is because it’s tempting to think that high gas scenarios are that way because they just have a big role for CCS, whereas low gas scenarios don’t. Figure S2 shows gas w/ and w/out CCS, so why not show that here? I think it would be good to.

We have added these subplots accordingly to (now) Figure 4.

Line 175-178 – This is a good point, and perhaps gets to the heart of why this analysis is very interesting but maybe not as novel (or maybe even useful) in its findings as it could be: it shows the outputs of IAMs, indicating (fairly intuitive) trends in how higher and lower gas (and indeed other fossil fuel) levels relate to other “levers” of decarbonisation. But it doesn’t have the ability to explain what drives the different results: particular assumptions from modellers, real-world constraints in roll-out rates of low-carbon technologies, some underlying assumptions which show (for example in the low-gas scenarios) that once gas is gone, then path-dependencies mean it won’t come back, versus no such assumptions in the high-gas scenarios? On the latter point, I’d be interested to see a comment from the authors on

whether they think it's realistic to see an (unabated, in many cases like transport) gas rebound later in the century, given the climate emergency we're in, and the pressure to shift away from fossil fuel industries.

As we stated above, while some of our results may not be “novel” or counter-intuitive, our primary aim with this paper is to provide a useful (we hope) and accessible explanation to a broad and interdisciplinary audience of researchers beyond the academic community directly engaged in developing or using the AR6 scenarios, as well as policymakers and non-academics, as to why some of the AR6 scenarios see gas phasing out whilst others see gas remaining a major part of the energy mix. Since there is such a large number of scenarios in the AR6 ensemble but relatively limited transparency around model/scenario assumptions (and inconsistent reporting of model outputs – for example, the discount rate is not reported by any model-scenario; see Table S3), and given that we were not involved with any of the specific underlying studies, we can only draw conclusions based on our own analyses and the literature. We think that our analyses have sufficiently captured key, distinguishing characteristics of different gas transition typologies. Please see our revised text on pages 8-9, including a comment on the plausibility of gas-rebound pathways.

Line 266 – Figure 4 is very nice, but made me realise that there is scope to do more with this paper, in terms of highlighting particular regions in which the feasibility challenges really become difficult. Could the authors, at least for the institutional indicator which I understand is in any case aggregated at the global level from underlying country level data, analyse where the particular institutional constraints / challenges are coming? This would help highlight where additional international funding / support / capacity building should be aimed. I think there's only really one comment on national pathways, around line 308, and more could be done here, even within the tight constraints of the 5-region IPCC database.

We agree that this is a very interesting question but unfortunately beyond the scope of our current analysis, which is focused on global pathways – we plan to delve into regional pathways and implications in a follow-up analysis. We have added some more text on regional implications based on other studies:

Page 14, line 41: “Finally, the pace and extent of the required global coal, oil, and gas reduction pathways (and their relative differences) will depend on many normative factors and policy choices, some of which cannot be adequately informed by IAMs alone⁵⁹. Many AR6-assessed scenarios that limit warming to 1.5°C with no or limited temperature overshoot show rapid and steep (70% or greater) reductions in coal, oil, and gas supply (and demand) between 2020 and 2050, enabled by high renewables integration, stringent assumptions about fossil CCS and CDR potential, and high carbon prices reflecting ambitious mitigation policies. Scenarios that see continued or increasing long-term gas supply are characterized by the opposing features, plus expectations of increasing gas demand as a transport fuel and for industry use. Thus, what is the level of risk society is willing to accept in terms of the probability of limiting long-term warming to 1.5°C? To what extent should we assume that future CDR and CCS can be developed and deployed at scale? What climate mitigation strategy might be the most feasible and/or the most desirable? Moreover, countries' differentiated capacities and circumstances, and other equity principles, should also be considered when disaggregating global reduction benchmarks into national pathways^{60–62}. Existing low-carbon scenarios, including those analyzed here, rarely incorporate equity and environmental justice considerations²⁷. For example, the rates of coal phase-out modelled by IAMs have been found to be much faster than historical precedents and would place the burden of stranded assets disproportionately on rapidly developing countries;

greater emissions reductions in the Global North and faster reductions in global oil and gas could allow for a slower coal phase-out in developing countries^{63,64}.”

Line 266 – Is the shaded area in Figure 4 the interquartile range?

We have revised the shaded range in this figure (now Figure 6) to show the full range of values and added this description to the legend, thank you for catching this omission.

#####

Reviewer #2 (Remarks to the Author):

This is an article that needs to be written (and should have been written years ago), but I feel this article needs a more solid analytical and statistical foundation. The article is primarily based on summary statistics of the AR6 scenarios database, with a little additional analysis, but I think a more robust statistical analysis is needed (for Nature Communications at least). There are two main issues associated with this: 1) It is stated that the AR6 scenarios database is unstructured, but nothing is done about it, and 2) There is a CART analysis, but it is hard to see how it informs the analysis. I have lots of comments below, but these are mainly suggestions for improvement and should be seen in the context of making this a stronger and more robust article. The comments are in order they appear in the article, not ranked in order of importance.

We are very appreciative of your careful review and constructive comments.

Longer comments:

1. The article says nothing about equity, but it would be expected that including equity considerations will change the global pathways. Since coal is now primarily used in non-OECD, greater equity will lead to slower reductions in coal and greater reductions in oil and gas (which is primarily in OECD). You can see this in some of the figures of Bauer et al 2020, mainly SI, <https://www.nature.com/articles/s41586-020-2982-5>. It is also rather intuitive. You could try and talk your way around this one, but it is sort of a fundamental issue. The analysis is based on the AR6 database, which includes almost no scenarios with equity considerations. Not only is the database unstructured, but it also misses some substantial and fundamentally important considerations. You need to be explicit about these shortcomings, and perhaps draw on relevant literature to make broader points.

Thank you for raising this important point. Due to scope and space constraints, we cannot do this topic sufficient justice here, but have nonetheless added the following text and citations:

Page 14, line 41: “Finally, the pace and extent of the required global coal, oil, and gas reduction pathways (and their relative differences) will depend on many normative factors and policy choices, some of which cannot be adequately informed by IAMs alone⁵⁹. Many AR6-assessed scenarios that limit warming to 1.5°C with no or limited temperature overshoot show rapid and steep (70% or greater) reductions in coal, oil, and gas supply (and demand) between 2020 and 2050, enabled by high renewables integration, stringent assumptions about fossil CCS and CDR potential, and high carbon prices reflecting ambitious mitigation policies. Scenarios that see continued or increasing long-term gas supply are characterized by the opposing features, plus expectations of increasing gas demand as a

transport fuel and for industry use. Thus, what is the level of risk society is willing to accept in terms of the probability of limiting long-term warming to 1.5°C? To what extent should we assume that future CDR and CCS can be developed and deployed at scale? What climate mitigation strategy might be the most feasible and/or the most desirable? Moreover, countries' differentiated capacities and circumstances, and other equity principles, should also be considered when disaggregating global reduction benchmarks into national pathways⁶⁰⁻⁶². Existing low-carbon scenarios, including those analyzed here, rarely incorporate equity and environmental justice considerations²⁷. For example, the rates of coal phase-out modelled by IAMs have been found to be much faster than historical precedents and would place the burden of stranded assets disproportionately on rapidly developing countries; greater emissions reductions in the Global North and faster reductions in global oil and gas could allow for a slower coal phase-out in developing countries^{63,64}."

Page 15, line 41: "While the scenarios assessed in the IPCC AR6 offer a wealth of information in terms of strategies for decarbonizing our energy systems, it is important for decisionmakers to understand their limitations and shortcomings, and the associated risks and ethical concerns of different modelled pathways, including the potential failure of CDR and CCS technologies to develop at scale^{65,77}, when using them to set global and national climate policy agendas. The scenarios' differentiated alignment with other important societal and environmental outcomes like the Sustainable Development Goals (SDGs) and equity should also be evaluated⁷⁸. Understanding the feasibility concerns of different scenarios can provide a valuable tool for identifying enabling conditions that can help to overcome them⁴⁵. We also note that our analysis reflects only the possibilities explored in the studies underlying the AR6 scenarios ensemble. Other researchers have, for example, modelled alternative "degrowth" scenarios in which continued growth in gross domestic production is not a prerequisite to support societal wellbeing, finding that such mitigation scenarios minimize many key feasibility and sustainability risks compared to technology-driven pathways⁷⁹."

2. The statement on lines 39 and 40 is factually correct, but misleading. As reference 8, Figures 3.1 and 3.2 show, the scenarios database is extremely skew to some models and projects. This skewness is even worse if one looks individually at C1, C2, and C3 separately. As Guivarch et al 2022 suggests, "several important aspects have to be addressed" when dealing with such a database, and I don't see that you have addressed these. Ok, AR6 Chapter 3 did not address these, and got away with it, but as a scientific community, we must do better. I think you need to have a solid discussion on how the bias in the database may affect your results. When you look at C1, are you showing me anymore than REMIND (41 of 97 scenarios). Or C3, 176 of 311 are from ENGAGE, with its very particular scenario design. To be clear, 1202 may be the number of scenarios and 44 may be the number of models, but that tells me nothing about the underlying statistical distribution, which largely renders summary statistics from the database unusable. Your challenge is to prove me wrong!

Thank you for raising another important point. You are right that we need to highlight this limitation of the AR6 scenario database more clearly throughout our paper, and factor in the influence of model and scenario protocols in our analyses of the drivers of different modeled levels of coal, oil, and gas supply. Having said that, the main purpose of our paper is to serve as an accessible summary for a broad and interdisciplinary audience of researchers beyond the academic community directly engaged in developing or using the AR6 scenarios, as well as policymakers and non-academics, as to what the AR6 scenarios say about fossil fuel transitions and why. As we are sure you know, there is still limited practice in and consensus on bias correction for over-/under-representation of climate mitigation scenario ensembles. And so, our approach is not to proceed with some additional treatment of the database (which could itself introduce other potential biases), but, rather, to be more comprehensive in

our analyses and discussions of the potential influence of model designs and scenario protocols on the outputs of the whole AR6 ensemble being analyzed. Consequently, we now factor in the model (version and/or family) and scenario project study in our CART, ANOVA, and gas cluster analyses, and draw out particular features of specific model families or scenario projects where appropriate.

As for summary statistics, we are now more careful to explain why we do show the median pathway in Figure 1 (i.e., for illustrative purposes, so that we can reference them later when discussing how different individual and CDR-limited pathways can be in terms of coal, oil, and gas reductions.). Moreover, we now show all of the individual pathways in Figure 1. (We elaborate on this more in response to your 4th comment below.)

3. Lines 52-54 are a little misleading, as you write in lines 400+. Chapter 3 Table 3.5 does not show total CDR, it was removed from the final version (see corrigenda), because the IAMers did not know what AFOLU or Afforestation represented in different models. In any case, your text gives the impression that DACCS is more prevalent than it is. You could perhaps bring some text from the methods forward here to be clear about this. As you write in the methods, all models have AFOLU, but roughly half afforestation, so really, models all implicitly include afforestation, we just don't know how much. All models, basically, include BECCS. DACCS is not so well represented. As for the previous comments, these numbers are biased by models too. If you remove one model from the database, maybe the statistics are rather different! This is problematic when you talk about the limited CDR pathways towards the end, where you are looking at a far more skewed version of the already very skewed database!

We took this specific wording verbatim from the IPCC AR6 WGIII page 3-7, i.e., “Pathways that likely limiting warming to 2°C or below involve some amount of CDR to compensate for residual GHG emissions remaining after substantial direct emissions reductions in all sectors and regions (high confidence)... **CDR options in the pathways are mostly limited to BECCS, afforestation and DACCS.** CDR through some measures in AFOLU can be maintained for decades but not in the very long term because these sinks will ultimately saturate (high confidence). {3.4}”

However, we note your concern that it can be misinterpreted and have revised our text accordingly.

Page 2, line 14: “The majority of low-carbon scenarios produced by IAMs rely extensively on CDR, mostly through bioenergy combined with carbon capture and storage (BECCS) and land sequestration; a few scenarios also employ direct air capture with carbon capture and storage (DACCS)^{19,22}.”

4. Lines 71+. Since this is not an IPCC report, I feel my duty to ask for justification of using median and interquartile range (IQR). What is your statistical justification? What about 0-100 (range), 5-95, 33-66, 25-75, median, mean, etc. Is there something about the statistical distribution which leads you to use IQR? You are not plotting half the data! Page 3 is all discussing the median. But what does the median even mean in this database? Why even plot Figure 1 in that style? You talk about % reductions in particular years, so maybe a more statistically prudent way of doing this is to show a histogram of time slices, which would somewhat avoid issues on choosing IQR or something else? What is the histogram of coal in 2050 (for example)? You also reference Guivarch et al, saying that the approach you have taken “has limitations”, but have you tried to address those limitations? We now know the database is unstructured and biased, so I don't think it is possible any longer to just brush that aside! You have to draw to address those limitations as best you can!

We want to focus on showing time series (rather than time slices which are mainly used in the IPCC reports) in this paper since there is growing policy and civil society interest in establishing Paris-aligned trajectories, reduction benchmarks, and even “end-dates” for coal, oil, and gas supply. That said, you are indeed correct that we need to be more careful in how we show and discuss results based on a median pathway (and a given statistical range). As we now clarify in the text, we show the median in Figure 1 (and now along with all individual, underlying scenarios) mainly for illustrative purposes to be able to show later how much individual or a subset of scenarios diverge from this “average” trajectory. We realize that this is an imperfect approach, but don’t see any other way of succinctly summarizing dozens to hundreds of different time series in the AR6 ensemble.

We now also show boxplot distributions of the coal, oil, and gas supply at key years in Supplemental Figure 1. We also test the sensitivity of the median pathways and IQRs of the full AR6 C1-C3 ensemble shown in Figure 1 to potential model bias:

Page 6, line 28: “Since model family is an important predictor of cumulative fossil fuel supply but is not equally represented in the AR6 ensemble (Figure S3), we test the sensitivity of the median pathways shown in Figure 1 to potential model bias. There is still limited practice in and consensus on bias correction for model over-/under-representation of climate mitigation scenario ensembles³⁶. Nevertheless, when we include only the scenarios with minimum and maximum 2020-2100 cumulative supply from each model family for a given fuel and category, the values and trajectories of the median pathways shown in Figure 1 do not change much, except for C1-gas in which the post-2050 values would be higher (Figure S12).”

5. Lines 199-120 you write “...higher gas supply also appear to rely on greater CDR...”. I have problems with “appear”. You should be able to show this? Perhaps this is a lead into the next section, but the next section does not seem to draw on the CART. CART is not mentioned until the methods, and I don’t really see how the analysis from lines 125+ is informed by the CART. Are the lines 126-130 the CART? That is not obvious. The split of gas into two parts is also a little arbitrary. All choices will be arbitrary, but since you use cumulative energy use to 2100, you could split the 2100 distribution 50-50. Or did you make the 2030 choice because that shows something special? Again, on Figure 2, not sure how helpful the IQR is, but I understand you have to make a choice. Basically, I am lacking the statistical justification for the choices you are taking.

On the CART analysis, we agree that the purpose of and insights drawn from this analysis need to be made clearer. We have revised the text quite substantially in this section, and moved the CART figures for C1 coal, oil, and gas from the Supplement into the main paper as Figure 2.

On the gas typology analysis, we have revised this analysis to be more statistically robust and applied k-means clustering to group the gas pathways into three different typologies based on their modelled values in 2030, 2050, 2075, and 2100. Please see our revised manuscript, figures, and methods for details. Similar to one of our responses to the first reviewer above, given that there is a large number of scenarios in the AR6 ensemble but relatively limited transparency around model/scenario assumptions (and inconsistent reporting of model outputs – for example, the discount rate is not reported by any model-scenario; see Table S3), and given that we were not involved with any of the specific underlying studies, we can only draw conclusions based on our own analyses and the literature. We think that our analyses have sufficiently captured key, distinguishing characteristics of different gas transition typologies.

6. Figure 3. I am going to have a lot to say about this figure... You may see bigger biases here, and model and project fingerprints. A key figure that is missing is total primary or final energy. I see you have components of demand, but you could probably drop those and just have PE or FE or both. That, I think, will explain quite a lot of your first order variation? It may be the top layer of your CART, if you restrict the variables? There are a few obvious fingerprints in the figures. The gas going down then up, that is MESSAGE. The high nuclear, that is MESSAGE and to a lesser degree GCAM. So high gas implies high nuclear? Why? Is there a causal reason for that, or is it just because MESSAGE is dominating that part of the distribution? Try taking MESSAGE out and see what the figures look like. Repeat for all models. Did you try total CCS (bio + fossil), why is fossil CCS so important but not BECCS? Transport demand gases, that is MESSAGE! You are likely to also find a quite strong ENGAGE imprint on this figure, as the experimental protocol of ENGAGE will cause the bifurcation you have in panel a. Basically, my concern here, is that you are just showing the outcomes of MESSAGE or ENGAGE. In all the figures I plot, nearly all the purple outliers show MESSAGE! That may have value, just like selecting an IMP might have value, but it might cloud your policy recommendations. If you select another variable or another Cx, then you will probably pick up a fingerprint of REMIND as it is so dominant, etc. The carbon price fingerprint, that may well be heavily affected by the ENGAGE model protocol, particularly in C2 and C3 where ENGAGE is more dominant. Etc. I think some effort is needed in Figure 3 to deal with model and project bias, as this is the only way to get to some of the most important policy conclusions, I think.

We now include total primary energy supply in our CART, ANOVA, and gas cluster-typology analyses, but do not find this to be an important predictor of coal, oil, gas supply variability.

To accompany our Figure 5 (formerly Figure 3), we now show the scenario project and model distributions across the different gas clusters in Figures S15-S18. Our revised analysis shows that C1 scenarios showing a gas “rebound” trajectory are generated by the following six models (and their associated project study): GCAM 5.3 (Project study: Ou 2021), GEM-E3 V2021 (ENGAGE), IMAGE 3.2 (van Vuuren 2021), MESSAGEix-GLOBIOM 1.1 (ENGAGE), MESSAGEix-GLOBIOM 1.2 (Kikstra 2021), and WITCH 5.0 (ENGAGE). Scenarios grouped into the “fast decline” and “slow decline” clusters belong to multiple projects, but are predominantly from the REMIND model family (see Figure S16).

In each of the C1-C3 categories, we find that scenarios from the ENGAGE project appear in all three gas clusters (Figure S15). The largest number of ENGAGE scenarios are found in the “rebound” cluster in C1, and in the “fast decline” or “decline” clusters in C2 and C3, respectively.

Thus, you are right that certain models dominate certain patterns of gas pathways (e.g., the REMIND model for C1 “fast decline” pathways). We now try to account for and show this influence (and highlight specific features of a given model family or scenario project where appropriate) but, again, the main focus of our analysis and discussion is to highlight and explain the model assumptions and scenario characteristics that are underlying different gas transition pathways. Please see our revised figures and text under the section “Characteristics of scenarios with different roles for gas” on pages 6-9.

7. I am a little uneasy on the IMP section. The IMPs are drawing out characteristics of one scenario, but it is unclear if that one scenario is more representative of the desired characteristics or the model or the intercomparison. For example, do all low demand scenarios have the same characteristics as IMP-LD, or is this heavily influenced by MESSAGE. There also seems to be a slight contradiction, in that you start off by analysing the database to get a spread across scenarios, but then fall back on to individual IMPs. You could go back another step, and do more stylised “if gas use is higher, then it needs more CCS, more CDR, or other fossil fuels to go down faster for a fixed carbon budget”. These are just random thoughts

on the IMPs, as to me it was not clear how they fit the overall narrative. In my view I think it is worth reflecting whether there is value in discussing the IMPs or expanding other parts of the analysis.

Given how large and diverse the AR6 scenario ensemble is, we think that there is value in focusing on a few individual scenarios in the database to convey how specific scenario storylines can influence coal, oil, and gas transition pathways, especially for non-experts. For example, what does the IMP-SP with its explicit focus on combining climate mitigation with sustainable development say about fossil fuel transitions? We are careful to emphasize clearly that the IMPs are illustrative, and not representative, of the AR6 scenario ensemble.

The narrative flow of our paper and analyses are as follows:

- 1) What do the AR6 scenarios that are consistent with limiting warming to well below 2C say about coal, oil, and gas supply, and why do the overall dependence differ? (CART analysis of cumulative 2020-2100 levels of coal, oil, gas supply.)
- 2) Why do we see such large variability in the long-term role for gas, especially from mid-century onwards? (ANOVA and cluster analysis of different gas typologies to explore the drivers and characteristics).
- 3) What are the implications of different, individual mitigation storylines (IMPs) and to taking a precautionary approach to CDR reliance? (Comparison of pathways under the IMPs and a CDR-constrained subset compared to the whole ensemble).
- 4) Discussion and policy implications of all our findings.

8. Line 369: “model itself is also an important predictor”. I think you need more justification on this point. If the model is an important predictor, how can you ignore it. There might be grounds, but you need to convince me of that.

As noted above, we now factor in the model (version and/or family) and scenario project study in our CART, ANOVA, and gas cluster analyses.

Minor comments:

9. Lines 33 and 34, references 4 and 5. Particularly reference 4, you would be much better off to use Friedlingstein et al 2022, Global Carbon Budget 2022, as it is a) peer reviewed, b) more recent than the IEA analysis. You could also use this for the statement on Ukraine, but the WEO may fit your needs better?

Thank you for your suggestion, we have changed this reference to Friedlingstein et al. 2022.

Page 1, line 37: “At the same time, global fossil fuel-derived carbon dioxide (CO₂) emissions rose to the highest level in history in 2021-2022⁵, and the ongoing global energy crisis has sparked a reappraisal of energy policies and priorities worldwide. It remains unclear whether this will serve to accelerate the clean energy transition or lock in fossil fuel dependence⁶, with many countries expanding fossil gas production or import capacity and others reviving coal use in their short-term responses⁷.”

5. Friedlingstein, P. et al. Global Carbon Budget 2022. *Earth System Science Data* **14**, 4811–4900 (2022).

6. Zakeri, B. et al. Pandemic, War, and Global Energy Transitions. *Energies* **15**, 6114 (2022).

7. IEA. *World Energy Outlook 2022*. <https://www.iea.org/reports/world-energy-outlook-2022> (2022).

10. Line 53, reference 11. Fine reference, but it is not based on AR6 which is your focus. Is there a relevant reference of AR6 showing CDR is not much different to SR15? Maybe just also add a ref to Ch3?

This was meant to be a generalized statement on IAMs. In any case, to avoid confusion, we now cite the IPCC AR6 WGIII here (it was the reference #8 at the end of this sentence), along with a new report on CDR.

Page 2, line 14: "The majority of low-carbon scenarios produced by IAMs rely extensively on CDR, mostly through bioenergy combined with carbon capture and storage (BECCS) and land sequestration; a few scenarios also employ direct air capture with carbon capture and storage (DACCS)^{19,22}."

19. Riahi, K. et al. Chapter 3: Mitigation pathways compatible with long-term goals. in *Climate Change 2022: Mitigation of Climate Change. Contribution of Working Group III to the Sixth Assessment Report of the Intergovernmental Panel on Climate Change* (Cambridge University Press, 2022).

22. Smith, S. et al. *State of Carbon Dioxide Removal - 1st Edition*. <https://osf.io/w3b4z/> (2023).

11. Lines 75. There is at least one study that would say many C1, C2, and C3 are not Paris compliant. <https://www.nature.com/articles/s43247-022-00467-w>

Thank you for raising this important point. We have revised our text and cited this reference to acknowledge this as follows:

Page 2, line 41: "Chapter 3 of the IPCC AR6 WGIII report vetted 1,686 global scenarios, of which 1,202 provided sufficient information for a temperature outcome categorization, ranging from "C1" to "C8"¹⁹. We interpret the C4-C8 scenarios to be inconsistent with the Paris Agreement's long-term temperature goal and focus on the C1-C3 categories with the lowest temperature outcomes: "C1" – limit warming to 1.5°C in 2100 with a likelihood greater than 50%, with no or limited overshoot; "C2" – limit warming to 1.5°C in 2100 with a likelihood greater than 50%, with high overshoot; and "C3" – limit peak warming to 2°C with a likelihood greater than 67%¹⁹. Here we analyze all scenarios in the C1-C3 ensemble and consider them to be relevant to the Paris Agreement's temperature limits, even if some individual scenarios may be considered to not be fully Paris-compliant depending on one's interpretation of its long-term temperature goal and of its other goals, as well as judgement on the probability, overshoot, and timing of the temperature change³². Of the 541 vetted C1-C3 scenarios, 94 C1, 131 C2, and 310 C3 scenarios report "primary energy supply" from coal, oil, and gas. Since these variables can include non-energy uses for fossil fuels, we interpret them as total supply for all uses (see Methods)."

12. Figure 1: I guess the vertical dotted lines sit on top of each other. Each figure should have three? Perhaps state that?

Yes, although to improve the readability of the figure we have decided to omit the vertical, dotted lines.

13. Lines 174+. But, you should be able to assess this?

Due to incomplete reporting of the relevant variables (and limited transparency around key model assumptions and parameters around path dependencies and technology roll-out rates), it is difficult for us to be able to draw definitive conclusions. For example, 11-13 out of 21 C1 scenarios showing a fast gas decline, and 6-15 out of 34 C1 scenarios showing a gas revival, report capital costs and capacity additions for various technologies. Nevertheless, we have elaborated on our findings for this section more. Please see our revised text and figures in the “Characteristics of scenarios with different roles for gas” section on pages 6-9.

14. Figure 4. If I understand correctly, you took all the criteria from Chapter 3, and implemented it?

Yes, we use the underlying feasibility assessment data of the IPCC AR6 provided by one of our co-authors, Elina Brutschin. To clarify, we have added the following to the text:

Page 12, line 32: “Figure 6 shows the feasibility assessments of the pathways plotted in Figure 5 (methods and details of the feasibility assessments of the AR6 scenarios are provided in the WGIII Annex II⁴⁸ and in Brutschin et al. 2021⁴⁵).”

15. Lines 275. Well, that is sort of a sweeping statement. Didn’t your figures, particularly if you look at the range (and not IQR), show that gas does not need to decline substantially and rapidly? In my view, coal, oil and gas do need to follow what is prescribed in your first sentence, but I do not think your analysis supports that (particularly for gas).

We have revised our discussion of these results to be more specific and accurate:

Page 3, line 22: “As shown in Figure 1, across the C1-C3 scenarios, global supply of coal and oil generally decline substantially and rapidly between now and mid-century, followed by a more gradual and variable reduction over time. More stringent temperature limits necessitate faster and greater reductions. Compared to coal and oil, there is less scenario consensus for the role of gas, with some seeing an almost complete phase-out by around 2050 while others see continued or increasing supply out to 2100.”

Page 14, line 2: “Our analysis finds that there is strong consensus in the AR6-assessed scenarios that global supply and demand of coal, oil, and gas need to decline substantially and rapidly between now and mid-century to limit warming to 1.5°C with limited overshoot, especially if CCS and CDR technologies fail to develop at scale.”

(For the latter statement, this is true even for gas in the majority of C1 scenarios, where in the gas “rebound” pathways, a phase-down out to mid-century is followed by a long-term increase.)

16. Line 282. Sure, fine, and likely true, but did your analysis show this? I don’t think you analysed methane? (hint, what happens if you put methane in your CART...)

We did analyze methane emissions (total and from energy) but did not show them in the main figures (we had included total methane emissions in Supplement figures S14-S16 in the submitted version). We now show methane emissions from the energy sector in the gas cluster-analysis results (Figures 4 and S13-S14), and still in the IMP and CDR-sensitivity analysis in Figure S19-S21. All C1-C3 scenarios show a strong phase-down of total and energy-based methane emissions after 2020. We have now added:

Page 14, line 35: “Secondly, other mitigation options – such as minimizing methane emissions from fossil fuel production processes – are important but are not a substitute for directly reducing fossil fuel supply in tandem with demand (see subplot m in Figures 4, S13-S14, and S19-S21).”

17. Line 290. 8600%. Sort of meaningless. This is small divided by small, right?

It’s a large number divided by a very small number. We want to convey the scale of how vast the “production gaps” are. We have now also added the absolute differences:

Page 14, line 12: “For example, compared to the median pathway modelled by all AR6-assessed C1 scenarios, global production levels derived from governments’ energy outlooks would lead to 1200% (145 EJ) more coal, 72% (92 EJ) more oil, and 135% (110 EJ) more gas in 2040. Under the CDR-limited C1 pathways, the respective “production gaps” increase to 8600% (156 EJ) for coal, 81% (98 EJ) for oil, and 350% (150 EJ) for gas.”

18. Line 303. “wind down”, I am fine with it, but considering this is an article about energy, I suspect many like me will read “wind (power)”. Perhaps use “reduce”...

Noted and changed to “reduce” here and elsewhere.

19. Overall, I think you could shorten the discussion. Not that you say anything wrong per se, but many statements are not necessarily supported by your analysis.

We have revised some of the text in our discussions and added supporting literature. We welcome further inputs on which specific statements you think are still not necessarily supported by our analysis (or, where relevant, supporting literature).

20. Line 348, footnote 1. Quite an important point. Most of the oil in 2050 and beyond is likely for feedstocks. Though, I suspect you can not find out the reporting quality. There is one non energy variable, perhaps in final energy, but I am not sure how many models report it.

We agree that this is an important detail. To clarify, we are interpreting the “primary energy supply” by coal, oil, gas variables to represent total supply for all intended uses (i.e., energy and non-energy uses), and this variable is the focus of our study. In trying to subsequently analyze which demand sectors may be heavily influencing this supply variable, we do account for non-energy uses to the extent possible. Unfortunately, the “Final energy|Non-Energy Use” variables are not consistently reported by the model-scenarios in the AR6 ensemble (please see the table below). However, some of this effect should be captured by the variables “Final Energy|Industry|xx”, which include use as feedstock for non-energy applications (though this variable also includes non-fossil sources). This variable is included in our statistical analyses and now shown in Figure S23, and we have added the following text to our Methods:

Page 16, line 18: “Global fossil fuel production values are taken from the “Primary Energy|Coal”, “Primary Energy|Oil”, and “Primary Energy|Gas” variables, which are provided in units of exajoules (EJ) per year, and reported by 97-100% of the C1-C3 scenarios (Table S3). These variables generally include “non-energy” uses of coal, oil, and gas (such as for chemical or plastics feedstocks), though this reporting varies between different models¹. Here we interpret the “primary energy supply” by coal, oil, gas

¹ Source: Volker Krey, IIASA, personal communication to Roberto Schaeffer, October 27, 2019.

variables to represent total supply for all intended uses (i.e., energy and non-energy uses). In trying to subsequently analyze which demand sectors may be heavily influencing this supply variable, we try to account for non-energy uses to the extent possible, using the mostly complete reported variable “Final Energy|Industry|xx” (where xx = Solids, Liquids, or Gases), although this variable does include non-fossil fuel sources. Figure S23 shows the individual, median, and interquartile range (IQR) 2010-2100 pathways, as well as boxplot distributions of the cumulative 2020-2100 values, of these variables as modelled by the C1-C3 scenarios.”

Variable	Number of scenarios reporting			Percent of scenarios reporting (%)		
	C1	C2	C3	C1	C2	C3
Final Energy Non-Energy Use Coal	35	78	152	36	58	49
Final Energy Non-Energy Use Oil	31	77	126	32	57	41
Final Energy Non-Energy Use Gas	31	77	124	32	57	40
Final Energy Industry Gases	84	123	289	87	93	93
Final Energy Industry Liquids	84	123	289	87	93	93
Final Energy Industry Solids	84	124	288	87	93	93
Final Energy (excl. feedstocks) Industry Gases	7	7	8	7	5	3
Final Energy (excl. feedstocks) Industry Liquids	7	7	8	7	5	3
Final Energy (excl. feedstocks) Industry Solids	7	7	8	7	5	3

21. Lines 368. “Carbon pricing”, within each category, like C1, C2, C3, the carbon pricing might be very heavily affected by the ENGAGE scenario protocol.

In the C1 category we see relatively high carbon prices in the SSP, Strefler, and other scenario projects. In the C2 and C3 categories they are primarily found in the ENGAGE and COMMIT scenarios. We now account for model and scenario project as explanatory variables in our CART (and dig more into the influence of carbon pricing in the next section and our gas-cluster analysis).

22. Line 365: CART. I did not really see how this was used to inform the analysis. There are a bunch of figures, but how did you use this knowledge in your analysis?

We agree that the purpose of and insights drawn from our CART analysis need to be made clearer. We have revised the text quite substantially in this section, and moved the CART figures for C1 coal, oil, and gas from the Supplement into the main paper as Figure 2.

23. Lines 405. C1, 4 out of 26 out of 97. So only 26 scenarios had sufficient data, and this is perhaps highly skew by model, and therefore highly biased. Likewise for C2 and C3. At least show how many scenarios did not have the data!

We are not sure what you are suggesting here, as we do already discuss in the text the number of scenarios that did not have the relevant data. It would not make sense (or be possible) to add the scenarios without data to the figure. Nevertheless, we have added a table to the Supplement that shows

the reporting completeness of variables used in our analysis, and identify the specific model-scenarios included in the 4 C1 and 6 C3 CDR-limited scenarios shown in Figures 5-6.

Supp Info

24. Line 8: AR6 assessment, not really the database?

Yes, good point. We have corrected the text to: *“The IPCC AR6 assessment of all submitted mitigation scenarios identified five different “illustrative mitigation pathways” (IMPs) that reflect different prominent mitigation strategies for reaching a given temperature outcome...”*

25. IMP-LD, Line 13. This scenario has high carbon price. Does that mean all LD scenarios have high carbon price? Or is this something about this scenario in MESSAGE? For the IMPs to be useful in this way, you need to be confident they are representative. I do not see that the Ch3 approach of selecting IMPs was so systematic, such as taking a CART and letting the IMPs fall out of the CART. I would be more convinced of the IMPs, if you showed that they were consistent with the CART, for example, or even if you had your own IMPs come out of your analysis.

In the preceding section, we conduct an analysis to identify common and representative features of scenarios with different gas supply typologies. In this section, our purpose is to demonstrate how much the coal, oil, and gas reduction pathways across the AR6 ensemble are influenced by different, individual scenario storylines and by CDR assumptions. Hence, we are using the IMPs for illustrative and not representative purposes here, and we are careful to discuss and interpret them that way. For example, the start of this section in the main text reads: *“To further assess and demonstrate how much the reduction pathways of coal, oil, and gas are influenced by different scenario storylines and assumptions, in this section, we first focus on a number of individual illustrative scenarios and then explore what happens when we take a conservative approach to future CDR potential. The AR6 scenario database identified five different “illustrative mitigation pathways” (IMPs) that reflect different prominent mitigation strategies for reaching a given temperature outcome (see Figure 5 legend). We emphasize that no two scenarios are alike or more “correct” than others; the IMPs are illustrative and not representative of the AR6 ensemble. The individual coal, oil, and gas supply pathways under the five IMPs, along with the median pathways across each of the C1-C3 scenarios, are shown in Figure 5.”*

Since the main purpose of our paper is to serve as an accessible summary on what the IPCC AR6 mitigation scenarios say about coal, oil, and gas, we would prefer to use the five IMPs identified in the AR6 WGIII for consistency, rather than identify new ones.

26. Figure S2. Mentions IQR in caption, but I don't see it in the figure.

Sorry about that. There seems to be a MacOS/Windows PDF conversion issue for the figures with shaded areas. We've now converted all relevant figures to JPEG files in the revised draft.

27. I am not an expert on CARTs, but I guess in context, you have to tell the system what variables to look at. You have used a mix of variables, some of which might be reported. Some will overlap, things like SE and PE, BE and BECCS, etc. To what degree is it important to decide what variables that CART can consider? But most importantly, what did I learn from the CART, and how is it used in the main paper. This I missed!

As we stated above, we agree that the purpose of and insights drawn from our CART analysis need to be made clearer. We have revised the text quite substantially in this section, and moved the CART figures for C1 coal, oil, and gas from the Supplement into the main paper as Figure 2.

As for the independent variables provided to CART, you are correct that one has to pre-select the variables, like in other regression techniques for prediction, but CART makes no assumptions about the relationships between the independent variables. The independent categorical and continuous variables we chose to include in our analysis reflect those that are frequently discussed in the literature and/or judged to likely influence the level of fossil fuel dependence, and are reported by more than two-thirds of the model-scenarios. These fall under five broad categories as follows: 1) Total primary energy supply and primary energy supply from alternative sources, including other fossil fuels; 2) End-use by different sectors; 3) Technologies that would enable more or less fossil fuel use, including CDR and CCS; 4) Carbon pricing – a general indicator of policy stringency and disruptiveness in the IAMs⁴¹; and 5) Model family and scenario project.

#####

Reviewer #3 (Remarks to the Author):

Many thanks to the authors for this paper. It provides some interesting insights into the prospects for fossil fuel demand under a range of different climate scenarios.

There are four main analytical elements to the paper as I see it: a high-level analysis of the IPCC scenario, a grouping of some of these scenarios according to gas use, a Classification and regression tree (CART) analysis, and feasibility assessments of the pathways.

The high-level analysis of the IPCC scenario database looks robust and I have not seen this published elsewhere. I was not able to follow fully the Classification and regression tree analysis (CART) and whether this has been carried out correctly as this is outside my area of expertise, but the results make initiative sense. I did not understand the feasibility assessments of the pathways (underpinning Figure 4), as these are not really described in any detail in the paper or the supplementary information. Much more description of what this is based on and where the results have come from is needed.

Thank you very much for taking the time to review our paper. Apologies for not having made clear where the feasibility assessment data derive from. We have now added this clarification:

Page 12, line 32: "Figure 6 shows the feasibility assessments of the pathways plotted in Figure 5 (methods and details of the feasibility assessments of the AR6 scenarios are provided in the WGIII Annex II⁴⁸ and in Brutschin et al. 2021⁴⁵)."

I am not convinced by the analysis on gas. In general, this section requires a much more balanced discussion on what are the results and a better justification for the conclusions. The authors conclude (in the abstract) that "gas remains a significant energy source only if its use can be coupled to carbon capture and storage, combined with relatively limited renewables deployment and extensive land-based carbon sequestration". It is not clear how these specific elements were selected: Figure 2 shows there are elements that appear to have a much larger impact on the role of gas. For example, differences in nuclear and bioenergy use are much larger between the two scenario groups are much larger than renewables. Scenarios with more gas, have much lower long-term CO2 price and these elements seems

much more noteworthy and significant than the ones highlighted by the authors. The conclusion on land-based carbon sequestration is especially weak as the figure shows that while they have more land-based carbon sequestration, they have much lower BECCS. The sum of BECCS and AFLOU (i.e. total carbon dioxide removal) does not look materially different between the two scenario groupings.

Given what is shown, there is a strong possibility that the differences between the gas groups are more a function of the IAMs that are contributing more or fewer scenarios to each scenario group rather than real differences in findings. For example, it may be that one model contributes the majority of scenarios in one group and this model has an inherent inclination to have higher energy demand, less gas and lower CO2 prices.

Due to the 150 word-limit for the abstract, we cannot list all the relevant drivers of gas variability, which were informed by a combination of our CART, gas pathway-grouping, and scenario-filtering (by CDR limits) analyses in our originally submitted abstract. (By “land-based sequestration”, we meant to refer to negative AFOLU emissions only in a concise way, but appreciate that this wording is too confusing.) In any case, we have now edited the results highlighted in this sentence in the abstract to be focused on and informed by our gas ANOVA and cluster analyses of the C1 pathways only for greater clarity.

As for our gas analyses, we have expanded our statistical analyses and associated discussions of the results, and also accounted for the influence of different models (or model families) and scenario projects in our CART, ANOVA, and cluster analyses. We describe the characteristics, dynamics, and implications of both the lower- and higher-gas scenarios. Please see our revised manuscript, figures, and Supplement for details, especially the “Characteristics of scenarios with different roles for gas” section on pages 6-9.

A second area that needs attention is the implications of their findings on the non-energy use of fossil fuels (e.g. the use of oil as a petrochemical feedstock). This is mentioned briefly in the methods section, but given the importance of the sector to fossil fuel demand, it needs to be analysed in much more detail.

We agree that this is an important detail. To clarify, we are interpreting the variables “primary energy supply” by coal, oil, and gas to represent total supply for all intended uses (i.e., energy and non-energy uses). This is because some model-scenarios do report non-energy usage under this variable, but this is not consistent across all model-scenarios. In trying to subsequently analyze which demand sectors may be heavily influencing this supply variable, we do account for non-energy uses to the extent possible. Unfortunately, the “Final energy|Non-Energy Use” variables are not consistently reported by the model-scenarios in the AR6 ensemble (please see the table below). However, some of this effect should be captured by the variables “Final Energy|Industry|xx”, which include use as feedstock for non-energy applications (though this variable also includes non-fossil sources). This variable is included in our statistical analyses.

	Number of scenarios reporting			Percent of scenarios reporting (%)		
	C1	C2	C3	C1	C2	C3
Final Energy Non-Energy Use Coal	35	78	152	36	58	49
Final Energy Non-Energy Use Oil	31	77	126	32	57	41
Final Energy Non-Energy Use Gas	31	77	124	32	57	40

Final Energy Industry Gases	84	123	289	87	93	93
Final Energy Industry Liquids	84	123	289	87	93	93
Final Energy Industry Solids	84	124	288	87	93	93
Final Energy (excl. feedstocks) Industry Gases	7	7	8	7	5	3
Final Energy (excl. feedstocks) Industry Liquids	7	7	8	7	5	3
Final Energy (excl. feedstocks) Industry Solids	7	7	8	7	5	3

Finally, the authors conclude that “governments need to urgently plan for and implement a global wind-down of coal, oil, and gas, starting now.” It is not clear how this conclusion is reached: it may well be the case that a wind down in fossil fuel use can be achieved by boosting clean energy technology deployment and fossil fuel supply falls as a consequence. While a plan for reducing fossil fuel use would be helpful to avoid negative impacts, the paper does not justify why implementing a wind down is necessary.

We have edited this policy recommendation to be less prescriptive and to expand on the accompanying justifications by adding text and supporting literature to explain why addressing both fossil supply and demand (in addition to other mitigation levers) can increase the effectiveness – and equity – of climate mitigation policies and actions. Please see our revised text on page 14.

In general, this could be an interesting paper, but I am worried that some of the findings and the conclusions suffer from selection bias.

We hope that our revisions and responses have sufficiently addressed your concerns and welcome further feedback on which specific findings/conclusions may still be problematic in your opinion.

REVIEWER COMMENTS

Reviewer #1 (Remarks to the Author):

I think the authors did a thorough job of bringing out the key points and caveats on the scenarios analysed, particularly around gas. Whilst there remain some unanswered (unanswerable) questions around some of the underlying drivers of the results across studies and scenarios, the paper is a useful addition to the literature on the role of fossil fuels in different pathways. I'd be happy to see it published in Nature Communications.

Reviewer #2 (Remarks to the Author):

Thankyou to the authors for an extensive revision of the paper. They have done a great job, but I still have some broad comments. I do not think this should stop publication, it is the editors call. Maybe some minor tweaks to the text are sufficient.

I broadly agree with the conclusions of the paper, but I am not sure the analysis convincingly shows that. The conclusions can come from simple heuristics. I think the authors have ultimately used the database to justify the heuristics (which I guess is ok, but should be stated?). I think the additional analysis of the authors sort of confirms this view, though, I accept the authors will disagree.

The simple heuristic approach would say that:

- Fossil fuel use (unabated) has to go down rapidly to remain within a carbon budget
- Using more CCS and CDR would allow more coal, oil, or gas
- Using more non-fossil energy would lead to less fossil energy (under fixed energy consumption, and with different amounts of CCS/CDR given the fixed carbon budget)

The scenarios show this, more or less, but it is not possible to quantify the statements above!

Low gas use would be associated with a smaller carbon budget, less CCS / CDR, and higher shares of total energy use met by energy carriers other than gas. The opposite would be true for high gas use. This is very similar to what the authors write in their abstract and throughout the paper. So to be clear, I don't disagree with the authors findings, I am less confident that the scenario database gives much confidence...

What does the database potentially add? It can add some more numbers, justification, relative differences, etc. Though, to quantify this is really hard from the database, because there is no experiment in the database that looks at addressing the questions posed by the authors. So, does the database have sufficient structure to actually answer the question posed by the study? This I don't know. The authors would say yes, but I am not convinced. To explain that is not a one-line statement, but more collection of examples and counter examples.

Hence, I just have a bunch of comments, which the authors may just take as comments and ignore. Or perhaps they can tweak the text a little to be sufficiently nuanced. I would not want to block publication based on my comments!

I think the abstract can be improved. This is probably the fault of us reviewers making suggestions. In some places you have qualitative comments, other places have quantitative, this is a bit inconsistent. The limited CDR is quantified by the scenarios without limits are not, this is screaming for a comparison. In such a paper, the "substantially" needs quantification. I think this just requires minor tweaks, and here is a suggestion with minimal changes to your original (though, take this is a suggestion, I am not trying to take over the paper!)

Scenarios limiting warming to 1.5°C with no or low overshoot (C1), global coal, oil, and fossil gas supply decline X%, Y%, and Z%, from 2020 to 2050, but the long-term role of gas thereafter is highly variable. "Fast decline" gas pathways are characterized by lower carbon capture and storage and carbon dioxide removal (CDR), higher carbon prices, higher renewables, and lower gas demand for transportation. Gas "rebound" pathways are characterized by the opposite features. Under these C1 scenarios with limited CDR, global coal, oil, and gas supply (and use) decline on average by 99%, 70%, and 84%, respectively, between 2020 and 2050, substantially faster than scenarios with no limits on CDR. Decision makers should consider the feasibility, desirability, and risks of different Paris-aligned pathways when setting policy agendas.

Following are a range of minor comments, mainly of examples and counter examples:

1. Line 21: "more stringent assumptions". Do you show this? CCS/CDR may be lower, yes, but are they lower because of more stringent assumptions or other model and scenario factors?
2. Line 22: In Figure 4g, renewables seem lower in the fast decline? What do you mean by "integration"? Surely you just mean renewables? Not that renewables are better integrated?
3. Line 63: "a few". I suspect "fewer" is more correct?
4. Figure 2, the CART.
 - a. Panel C. PE/Gas/CCS is repeated on the right? I guess that is possible in CART, but not sure what it means?
 - b. I am not completely sure the project study makes sense in the CART, I will have some more comments on that below.
5. Line 150: Reaching median values less than 10?
6. Figure 3. I think this is a pretty important figure. It also makes it challenging to be overly confident with your conclusions (other than heuristics). Model and project explain nearly all the variance. Fossil CCS is important, yes, but model and scenario are far more important. If model and scenarios are important, than you have to be confident you have a sufficient statistical sample of the scenario space.
7. Line 207+: There are also a bunch of scenarios that have a continual increase in gas use, at least when I plot them. They dont seem to fall into 1, 2, or 3. Looking at Figure 4a, I would suspect that many of the scenarios in the group 3 are actually increasingly in gas. If I look at GCAM and IMAGE, eg, they are not really of type 1, 2, or 3? Does this mean that some of the features of these scenarios are captured in group 3?
8. Line 240: "a sustained increase in renewables integration". (not sure what integration means). The figure seems to suggest that renewables grow in all, but that it grows less in the rapid decline scenario? Maybe 200EJ less? This may be due to lower PE? This did not come up in the CART, as you start the CART at coal, oil, or gas, not at PE.
9. Line 250. More positive assumptions on RE learning rates and costs. I could not see that in any of the figures, except offshore wind. Is the analysis showing this, or are these just very easy to justify heuristics?
10. Line 252: More stringent assumptions on CCS? Or you mean lower CCS? If the assumptions are more stringent it is a different question
11. Line 256: How many of these are MESSAGE though? Looking at the scenarios, it looks like this is mainly MESSAGE? Either way, this is a part of the problem of the study, via the database. If MESSAGE does dominate those scenarios, it means that message has high AFOLU and el gas, not that a gas rebound has those characteristics. These are sort of fundamental points if you ask me. Do all scenarios with high AFOLU and high el gas also have gas rebounds, or just MESSAGE?
12. Line 264. I am not sure I read much into the project study variable. First, it is empty for many studies! Second, some of these are very specific studies. Ou 2021 is a sensitivity study on MAC curves. Just plot the gas in C1, C2, C3. I am not sure I would want to use those scenario to make conclusions about the future of gas. I think if you plot CO2, it is a straight line to zero in 2030 or 2040 or something. These are sensitivity studies.
13. ENGAGE. There are two sets of mitigation scenarios, the not below zero and the standard ones. You really would need to split these out as separate studies I think. The standard ones have many models with a return to gas, mainly WITCH. In the not below zero, several other scenarios start to

have a return to gas, such as MESSAGE, POLES, COFFEE. There are other issues with the ENGAGE scenario design, I would argue, but I am not sure these are scenarios suitable for playing out the differences in low high CCS / CDR, etc. You could make the case if you removed the not below zero scenarios, though. Perhaps. It is something you really have to work through scenario by scenario, to see what makes sense, and what is biasing the sample.

14. Line 283: "seems implausible in our perspective". Well, I agree with you 100%. But this is obviously not what the modellers think, and they are supposedly the experts. The IPCC assessment implicitly also endorses it by not calling it out. So what is the justification for it to be implausible? I am not sure I have an answer to this myself, but how is it possible to justify your statement? The fact that you have to find possible explanations is an important point in itself!

15. Line 291. Sure, but the gas rebound also happens in scenarios that can have CO2 go below zero

16. Line 397: "especially", perhaps "more so"

17. Line 458: "CCS... below those modelled mitigation pathways", well, yes, but this can easily be explained by the fact that those pathways have a huge global carbon price and perfect information, reality has neither of those. If there was a 1000\$/tCO2 tax, maybe CCS would follow the scenarios? Who knows...

Reviewer #3 (Remarks to the Author):

Many thanks to the authors for the revisions. They seem to have addressed most of the comments I and the other reviewers raised with them.

One exception is on non-energy uses of fossil fuels. The reason given for not commenting further on this is that these data do not exist in the IPCC scenario database. The authors indicate that "Final Energy|Industry|xx" is used as a proxy for non-energy use, but this is not really appropriate (only around 15% of final energy use by industry is for feedstocks). This is a failure mostly of the IPCC scenarios not reporting these data, but I feel that the conclusions the authors make on the reductions necessary in fossil fuel use need to be heavily caveated to indicate that their analysis – and most IPCC scenarios – do not properly account for non-energy use of fossil fuels and that this has large implications for the rates of the reductions in fossil fuel use consistent with climate-constrained scenarios.

The new methods the authors have introduced to the paper (e.g. the ANOVA, and gas cluster analyses) are beyond my area of expertise. I am unable to comment on whether they are appropriate to use and/or have been carried out correctly processes, although the results make intuitive sense.

As mentioned before, my main concern with the paper is that the main analysis in the paper is quite simple and results are broadly to be expected (they have also been broadly covered in other recent papers e.g. <https://www.nature.com/articles/s41558-022-01576-2>) . It therefore remains an editorial decision as to whether this warrants publication.

RESPONSE TO REVIEWER COMMENTS

We would like to sincerely thank all three referees again for taking the time to comprehensively review our revised manuscript. Below, we respond to the comments in detail. The referees' comments are in plain text, our responses are in blue text, and changes to the manuscript are *in italics*. In our responses we refer to page and line numbers according to our revised manuscript ("03_AR6-FF_revised-June2023.docx").

#####

Reviewer #1 (Remarks to the Author):

I think the authors did a thorough job of bringing out the key points and caveats on the scenarios analysed, particularly around gas. Whilst there remain some unanswered (unanswerable) questions around some of the underlying drivers of the results across studies and scenarios, the paper is a useful addition to the literature on the role of fossil fuels in different pathways. I'd be happy to see it published in Nature Communications.

Thank you very much for your positive review of our revised manuscript.

#####

Reviewer #2 (Remarks to the Author):

Thank you to the authors for an extensive revision of the paper. They have done a great job, but I still have some broad comments. I do not think this should stop publication, it is the editors call. Maybe some minor tweaks to the text are sufficient.

I broadly agree with the conclusions of the paper, but I am not sure the analysis convincingly shows that. The conclusions can come from simple heuristics. I think the authors have ultimately used the database to justify the heuristics (which I guess is ok, but should be stated?). I think the additional analysis of the authors sort of confirms this view, though, I accept the authors will disagree.

The simple heuristic approach would say that:

- Fossil fuel use (unabated) has to go down rapidly to remain within a carbon budget
- Using more CCS and CDR would allow more coal, oil, or gas
- Using more non-fossil energy would lead to less fossil energy (under fixed energy consumption, and with different amounts of CCS/CDR given the fixed carbon budget)

The scenarios show this, more or less, but it is not possible to quantify the statements above!

Low gas use would be associated with a smaller carbon budget, less CCS / CDR, and higher shares of total energy use met by energy carriers other than gas. The opposite would be true for high gas use. This is very similar to what the authors write in their abstract and throughout the paper. So to be clear, I don't disagree with the authors findings, I am less confident that the scenario database gives much confidence...

What does the database potentially add? It can add some more numbers, justification, relative differences, etc. Though, to quantify this is really hard from the database, because there is no experiment in the database that looks at addressing the questions posed by the authors. So, does the database have sufficient structure to actually answer the question posed by the study? This I don't know. The authors would say yes, but I am not convinced. To explain that is not a one-line statement, but more collection of examples and counter examples.

Hence, I just have a bunch of comments, which the authors may just take as comments and ignore. Or perhaps they can tweak the text a little to be sufficiently nuanced. I would not want to block publication based on my comments!

Thank you very much for your careful review of our revised manuscript, and for sharing your thoughtful perspective and suggestions. We agree that the general logic you outlined in the bullet points above can be readily deduced without the use of an IAM and is well understood in the climate mitigation modelling community. However, our intent (as explained in the last paragraph of our introduction section) is not to use the AR6-assessed scenarios to justify this simple logic. Rather, it is to explore what the IAMs do say about the pace and timing of fossil fuel reductions under different carbon budgets and why, especially for fossil gas, as well as to discuss the risks and limitations of following different mitigation roadmaps (including those with different feasibility characteristics) for academic and non-academic audiences. In a sense, we are aiming to provide more texture, and more detailed results, concerning fossil fuels than were presented in AR6 WGIII (our abstract details these aims more clearly) and, even if that is a fairly modest aim, we believe it is a very important one and one that we have succeeded in.

I think the abstract can be improved. This is probably the fault of us reviewers making suggestions. In some places you have qualitative comments, other places have quantitative, this is a bit inconsistent. The limited CDR is quantified by the scenarios without limits are not, this is screaming for a comparison. In such a paper, the "substantially" needs quantification. I think this just requires minor tweaks, and here is a suggestion with minimal changes to your original (though, take this is a suggestion, I am not trying to take over the paper!)

Scenarios limiting warming to 1.5°C with no or low overshoot (C1), global coal, oil, and fossil gas supply decline X%, Y%, and Z%, from 2020 to 2050, but the long-term role of gas thereafter is highly variable. "Fast decline" gas pathways are characterized by lower carbon capture and storage and carbon dioxide removal (CDR), higher carbon prices, higher renewables, and lower gas demand for transportation. Gas "rebound" pathways are characterized by the opposite features. Under these C1 scenarios with limited CDR, global coal, oil, and gas supply (and use) decline on average by 99%, 70%, and 84%, respectively, between 2020 and 2050, substantially faster than scenarios with no limits on CDR. Decision makers should consider the feasibility, desirability, and risks of different Paris-aligned pathways when setting policy agendas.

Thank you for your helpful suggestion. We were concerned about exceeding the 150-word limit too much. But, if the editor is ok with a slightly longer abstract, we would like to edit the abstract accordingly to the following:

"The Intergovernmental Panel on Climate Change Sixth Assessment Report's scenario database is an important resource for informing policymaking on energy transitions. However, there is a large variety of models, scenario designs, and resulting outputs. Here we analyze what the scenarios consistent with limiting warming to well below 2°C say about the speed, trajectory, and feasibility of different fossil fuel

reduction pathways and why. For example, in “C1” scenarios limiting warming to 1.5°C with limited overshoot, global coal, oil, and fossil gas supply (for all uses) decline on average by 95%, 62%, and 42%, respectively, from 2020 to 2050, but the role of gas thereafter is highly variable. High-gas pathways are enabled by high carbon capture and storage (CCS) and carbon dioxide removal (CDR) reliance, but are likely associated with inadequate representation of regional CO₂ storage capacity, technology adoption and path dependencies, and a potential carbon price modelling artefact in certain models and scenario designs. If CDR and fossil-CCS are constrained by potential real-world limits, the respective coal, oil, and gas reductions modelled become 99%, 70%, and 84%. Our analysis supports the need for policymakers to adopt unambiguous near- and long-term reduction benchmarks in global coal, oil, and gas production and use alongside other climate mitigation strategies.”

Following are a range of minor comments, mainly of examples and counter examples:

1. Line 21: “more stringent assumptions”. Do you show this? CCS/CDR may be lower, yes, but are they lower because of more stringent assumptions or other model and scenario factors?
2. Line 22: In Figure 4g, renewables seem lower in the fast decline? What do you mean by “integration”? Surely you just mean renewables? Not that renewables are better integrated?

1. We have now updated this text in the abstract following further analyses of the individual scenarios in the gas fast-decline versus rebound pathways (please see the updated text in this section, along with Table S4 and Figures S17-S19 in the Supplement).

3. Line 63: “a few”. I suspect “fewer” is more correct?

We think “a few” reads more correctly here but leave this to the editorial team: “The majority of low-carbon scenarios produced by IAMs rely extensively on CDR, mostly through bioenergy combined with carbon capture and storage (BECCS) and land sequestration; a few scenarios also employ direct air capture with carbon capture and storage (DACCS).”

4. Figure 2, the CART.

a. Panel C. PE/Gas/CCS is repeated on the right? I guess that is possible in CART, but not sure what it means?

Yes, this is possible in CART because the branching can occur at different thresholds of the same variable.

b. I am not completely sure the project study makes sense in the CART, I will have some more comments on that below.

We are using the project study variable to broadly capture large differences in scenario design, but appreciate that this is imperfect. Given that there is a large number of individual projects and models (Figure S3), we now aggregate some of the individual scenario projects into one “Others” category, and use these aggregated “Scenario project” and “Model family” variables in our updated CART and ANOVA analyses, so that the variance analyses do not simply reflect differences in individual scenarios. We have also added further details and a caveat in the methods section:

Page 17, line 18. *“The independent categorical and continuous variables we chose to include in our analysis reflect those that are frequently discussed in the literature^{19,21,36} and/or judged to likely influence the level of fossil fuel dependence, and are reported by more than two-thirds of the model-scenarios. These fall under five broad categories as follows: 1) Total primary energy supply and primary energy supply from alternative sources, including other fossil fuels; 2) End-use by different sectors (for industry use, the available variable accounts for non-energy uses but includes both fossil fuel- and bio-based*

sources); 3) Technologies that would enable more or less fossil fuel use, including CDR and CCS; 4) Carbon pricing – a general indicator of policy stringency and disruptiveness in the IAMs⁴⁰; and 5) Model family and scenario project (Given the large number of individual project studies in the AR6 ensemble, the latter variable is created from the “Project_study” metadata variable by grouping projects with a relatively small number of scenarios (Figure S3) into one “Others” category. We use this as a proxy for capturing broad differences in scenario design, but note that differences can also exist within a large project, such as ENGAGE which specifically compares net-zero budget against end-of-century budget designs⁴⁰)...”

5. Line 150: Reaching median values less than 10?

Yes, we had said “global coal supply on average...” but to clarify this more, the text is edited to: “Across the C1-C3 scenarios, global coal supply on average peaks around 2015 (Figure S2) and rapidly declines (the fastest of all three fuels), with the median pathway reaching values below 10 EJ/yr in the C1 scenarios, and below 35 EJ/yr in the C2-C3 scenarios, after 2040 (Figures 1a-c).”

6. Figure 3. I think this is a pretty important figure. It also makes it challenging to be overly confident with your conclusions (other than heuristics). Model and project explain nearly all the variance. Fossil CCS is important, yes, but model and scenario are far more important. If model and scenarios are important, than you have to be confident you have a sufficient statistical sample of the scenario space. Yes, indeed model and scenario design are important drivers of the gas variance, and we explore the underlying reasons in the gas-cluster analyses and discussions that follow directly (please see our expanded analysis and updated text there). Since the AR6 scenario database is an unstructured ensemble, it is beyond our control to ensure sufficient statistical representativeness of each model and scenario project; we are trying to analyze the AR6 ensemble as is. In addition, we also explore other questions in this paper that do not require a statistical sample -- for example, what do individual, illustrative scenarios say about different fossil fuel reduction roadmaps?

7. Line 207+: There are also a bunch of scenarios that have a continual increase in gas use, at least when I plot them. They dont seem to fall into 1, 2, or 3. Looking at Figure 4a, I would suspect that many of the scenarios in the group 3 are actually increasing in gas. If I look at GCAM and IMAGE, eg, they are not really of type 1, 2, or 3? Does this mean that some of the features of these scenarios are captured in group 3?

Yes, we now show individual pathways for C1 gas-rebound in Figures S17-S19. As you pointed out, the IMAGE and GCAM high-gas scenarios do not strictly follow the “rebound” typology of this cluster, but since some of them do show some sort of minor near-term decrease, they get grouped into the rebound cluster. This clustering method allows us to explore and summarize patterns across the C1-C3 ensemble, as it is not possible to provide a detailed analysis of each individual scenario. However, we do note in the text that a continually increasing typology also exists in several instances in the text.

8. Line 240: “a sustained increase in renewables integration”. (not sure what integration means). The figure seems to suggest that renewables grow in all, but that it grows less in the rapid decline scenario? Maybe 200EJ less? This may be due to lower PE? This did not come up in the CART, as you start the CART at coal, oil, or gas, not at PE.

All our analyses, including the CART and Figure 4, are focused on the “Primary Energy|Coal”, “Primary Energy|Oil”, and “Primary Energy|Gas” variables. By renewable integration (see, e.g., <https://www.iea.org/topics/renewable-integration>), we are referring to the capacity addition pathways (Figures 4o-r for C1 and S13o-r for C2). For clarity, we have edited the text to: “...sustained increases in renewable capacity additions from 2020 onwards.”

9. Line 250. More positive assumptions on RE learning rates and costs. I could not see that in any of the figures, except offshore wind. Is the analysis showing this, or are these just very easy to justify heuristics?

10. Line 252: More stringent assumptions on CCS? Or you mean lower CCS? If the assumptions are more stringent it is a different question

These two observations were partly informed by our own analyses of the capital costs and by relevant information from the literature. We have further expanded this analysis and the corresponding descriptions in the main text and Supplement to substantiate our findings and conclusions here. Please see the updated text on pages 8-10.

11. Line 256: How many of these are MESSAGE though? Looking at the scenarios, it looks like this is mainly MESSAGE? Either way, this is a part of the problem of the study, via the database. If MESSAGE does dominate those scenarios, it means that message has high AFOLU and el gas, not that a gas rebound has those characteristics. These are sort of fundamental points if you ask me. Do all scenarios with high AFOLU and high el gas also have gas rebounds, or just MESSAGE?

As we show in (now) Figure S15, 14 out of 34 scenarios in the C1 gas-rebound cluster are from MESSAGE. One scenario from MESSAGE appears in the “fast decline” and five in the “slow decline” clusters. As shown in the plot below, we find that the majority of gas-rebound C1 scenarios are associated with relatively large negative AFOLU CO2 emissions. We first describe broad patterns observed for the high-gas scenarios (by saying “The higher-gas “rebound” and “increase” C1-C3 scenarios are generally associated with one or more of the following features:”), before providing a more detailed description for specific groups of scenarios (.e.g, by model family) afterwards. Please also see Figures S17-S19.

12. Line 264. I am not sure I read much into the project study variable. First, it is empty for many studies! Second, some of these are very specific studies. Ou 2021 is a sensitivity study on MAC curves. Just plot the gas in C1, C2, C3. I am not sure I would want to use those scenario to make conclusions about the future of gas. I think if you plot CO2, it is a straight line to zero in 2030 or 2040 or something.

These are sensitivity studies.

We are not sure what you mean about the project study variable being “empty for many studies”. If you look at the AR6 v1.1 metadata file, column L “Project_study” is filled in for every single C1, C2, and C3 scenario. We are using this variable as a proxy to broadly capture top-level differences in scenario design.

We appreciate you pointing out that the Ou 2021 scenarios are from a sensitivity study (since this information is not readily provided in the AR6 metadata file) and agree that this is another limitation of the AR6 database. Indeed, one of the findings and conclusions we try to emphasize in our paper is that there is a huge diversity of modelled pathways within the AR6 ensemble, and it is important to understand what is driving differences in modelled outputs such as gas pathways, in order to use them to inform policies. In general, sensitivity scenarios are problematic if they substantially depart from other trends but in this case, the trajectories for the modelled gas pathways (and other variables) from Ou 2021 (GCAM model) do not significantly depart from other C1 gas “rebound” scenarios (please see Figures S17-S19). Therefore, we now note that the Ou 2021 scenarios are from a sensitivity study, but still keep them in our analysis of the AR6 ensemble.

Supplemental file, page 28: Figure S17. *Individual pathways of select variables for the C1 scenarios grouped into the “rebound (rb)” or “fast decline (fd)” gas clusters for select model families (see Figure S15). (We note that the GCAM scenarios from the Ou et al. 2021 study were from a sensitivity study, but since the modelled outputs for many variables, including gas supply, show similar trends to other C1 “rebound” scenarios, we include them in our analysis here.)*

13. ENGAGE. There are two sets of mitigation scenarios, the not below zero and the standard ones. You really would need to split these out as separate studies I think. The standard ones have many models with a return to gas, mainly WITCH. In the not below zero, several other scenarios start to have a return to gas, such as MESSAGE, POLES, COFFEE. There are other issues with the ENGAGE scenario design, I would argue, but I am not sure these are scenarios suitable for playing out the differences in low high CCS / CDR, etc. You could make the case if you removed the not below zero scenarios, though. Perhaps. It is something you really have to work through scenario by scenario, to see what makes sense, and what is biasing the sample.

Yes, we had tried looking at differences between the end-of-century (“full”) versus net-zero (“peak”) budget scenarios under the ENGAGE (and other) project studies. For example, below are figures comparing the full versus peak scenarios for carbon budgets of 500 and 1000 Gt. The majority of “peak” budget scenarios, except for those from REMIND and TIAM, show a gas rebound. However, some “full” budget scenarios also show a gas rebound.

We have further edited the text to clarify that the modelling artefact we postulate may only be associated with certain scenario designs and models: *“This gas revival pattern, also seen in some of the scenarios from the GEM-E3 and WITCH models, may be partly driven by a potential modelling artefact associated with the “net-zero-budget” scenario design from the ENGAGE project⁴¹, in which carbon prices initially increase but then decline after net-zero CO₂ emissions are reached around mid-century (Figure S17).”*

14. Line 283: “seems implausible in our perspective”. Well, I agree with you 100%. But this is obviously not what the modellers think, and they are supposedly the experts. The IPCC assessment implicitly also endorses it by not calling it out. So what is the justification for it to be implausible? I am not sure I have

an answer to this myself, but how is it possible to justify your statement? The fact that you have to find possible explanations is an important point in itself!

We now provide further justifications here for why a gas-rebound pathway may be implausible given our expanded analysis of the underlying IAM assumptions and relevant literature:

Page 10, line 1. “Nevertheless, given technology path dependencies (a common phenomenon in socioeconomic systems, which arises when initial conditions and their historical antecedents matter for eventual outcomes)⁴³, combined with the urgent need to rapidly decarbonize our energy systems, a gas phase-down followed by a revival seems, if not outright implausible, to at least require a careful justification that we could not find in the underlying studies. Due to the high heterogeneity in and limited transparency around IAM assumptions, combined with inconsistencies in variable reporting, we could not determine which model parameters and assumptions are specifically leading to this outcome. We offer three most likely possible explanations: (1) inadequate representation of real-world constraints on fossil fuel- and renewable-based technology innovation, adoption, diffusion, and phase-in/-out path dependencies⁴³ (for example, an overestimation of the costs of renewable technologies is a problem that has been found in many IAMs^{44,45} and which leads to higher mitigation value of CCS in electricity and hydrogen production⁴⁶); (2) overly optimistic assumptions or insufficient constraints on CCS and CDR potential (for example, only the REMIND model imposes constraints on the CO₂ injection rate and regional storage potential, which influence fossil-CCS, BECCS, and DACCS availability)^{29,47}; and, in the case of “net-zero-budget” scenario designs specifically, (3) a potential artefact resulting in carbon prices dropping around mid-century after net-zero CO₂ emissions are reached in some of the models. In light of the important policy implications, we urge the model-scenario developers to be more explicit about key parameters and assumptions and to critically examine the real-world implications of their modelled results for gas pathways.”

15. Line 291. Sure, but the gas rebound also happens in scenarios that can have CO₂ go below zero
Yes, that is why we have offered two other possible reasons here and stated that #3 only applies in the case of net-zero budget scenarios.

16. Line 397: “especially”, perhaps “more so”
Edited to “, and even more so”.

17. Line 458: “CCS... below those modelled mitigation pathways”, well, yes, but this can easily be explained by the fact that those pathways have a huge global carbon price and perfect information, reality has neither of those. If there was a 1000\$/tCO₂ tax, maybe CCS would follow the scenarios? Who knows...

We have added text and a reference to explain that many models are not imposing sufficient modelling constraints on CO₂ storage potential: “Global rates of CCS deployment continue to fall below expectations and remain far below those modelled in the mitigation scenarios, with a total annual capacity of 45 MtCO₂ as of 2021^{30,76}. IAMs generally assume that CO₂ storage is a low-cost and globally ubiquitous resource; however, Grant et al. showed that this may lead models to substantially overestimate the role of CCS (coupled to fossil fuel and bioenergy use and/or direct air capture) – while under-utilizing renewable deployment – for decarbonization, when accounting for the technical, financial, and institutional barriers that may impose practicable limits on regional injection rates²⁹.”

#####

Reviewer #3 (Remarks to the Author):

Many thanks to the authors for the revisions. They seem to have addressed most of the comments I and the other reviewers raised with them.

One exception is on non-energy uses of fossil fuels. The reason given for not commenting further on this is that these data do not exist in the IPCC scenario database. The authors indicate that “Final Energy|Industry|xx” is used as a proxy for non-energy use, but this is not really appropriate (only around 15% of final energy use by industry is for feedstocks). This is a failure mostly of the IPCC scenarios not reporting these data, but I feel that the conclusions the authors make on the reductions necessary in fossil fuel use need to be heavily caveated to indicate that their analysis – and most IPCC scenarios – do not properly account for non-energy use of fossil fuels and that this has large implications for the rates of the reductions in fossil fuel use consistent with climate-constrained scenarios.

Thank you for your comments. Please note, as stated in our Methods section, that the IAMs from which we are analyzing the C1-C3 scenarios (Figure S3) *do** typically account for non-energy use of fossil fuels in their reporting of “Primary Energy|xx”, even if the reporting of “Final Energy|Non-Energy Use|xx” is sparse. Therefore, we have more confidence in our interpretation of the implied fossil fuel reduction pathways, but less confidence in the drivers of these pathways, given that the non-energy use portion is not consistently reported. We can confirm this for all but one C3 scenario from the EPPA model.

*This is based on our knowledge of the COFFEE model and our email communications with Volker Krey (IIASA) in late 2019, in which he said the following: “For most IAMs represented in IAMC [the SR1.5 database] the answer is that non-energy use of energy carriers is included in the primary energy reporting. As part of the EMF27 and AMPERE projects back in 2013 we ran a survey that, among other things, asked this question and **the following models confirmed that non-energy use is included in their primary energy reporting: AIM/CGE, AIM/Enduse, BET, DNE21+, GCAM, GEM-E3, IMAGE, MERGE-ETL, MESSAGE, POLES, REMIND, TIAM-World, WITCH.** Looking at the SR1.5 scenarios, the following five models were not included: C-ROADS, GENESyS-MOD, IEA WEM, IEA ETP, Shell. For the IEA models I am pretty sure they include non-energy use as well, for the other three you would need to check independently.”

To further clarify this issue and caveat our analysis accordingly, we have edited and added text as follows:

Page 3, line 7: “Since all but one C3 scenario analyzed in this study are generated from IAMs that typically include non-energy uses of fossil fuels under their reporting of the “Primary Energy|xx” variable (see Methods for details), we broadly interpret this variable to represent total supply intended for all combustion and non-combustion uses. However, the completeness with which non-combustion uses are accounted for varies between different models and could have significant implications for the resulting fossil fuel reduction pathways under a given carbon budget.”

Page 16, line 31 (Methods): “Global fossil fuel supply values are taken from the “Primary Energy|Coal”, “Primary Energy|Oil”, and “Primary Energy|Gas” variables, which are provided in units of exajoules (EJ) per year, and reported by 97-100% of the C1-C3 scenarios (Table S3). According to our knowledge of the COFFEE model and a 2013 survey ran by the International Institute for Applied Systems Analysis (IIASA) (V. Krey, IIASA, personal communication to R. Schaeffer, October 27, 2019), 10 of the 11 model families whose scenarios (534 out of 535 scenarios) are analyzed in this paper include non-energy use under their reporting of “Primary Energy|xx”: AIM, COFFEE, GCAM, GEM-E3, IMAGE, MESSAGE, POLES, REMIND,

TIAM, and WITCH. (We do not know whether the EPPA model accounts for non-energy use, but there is only one C3 scenario from EPPA analyzed here.) Thus, here we broadly interpret the “Primary Energy|xx” variable to represent total fossil fuel supply for all intended uses (i.e., combustion and non-combustion uses, such as for chemical or plastics feedstocks). However, the level of detail to which non-combustion uses are accounted for likely varies between different models. More consistent documentation and reporting of this issue, including by the “Final Energy|Non-Energy Use|xx” output variables (currently reported by 32%-58% of the C1-C3 scenarios), would aid our analysis and interpretation. In trying to subsequently analyze which demand sectors may be heavily influencing modelled fossil fuel supply, we try to account for non-energy uses to the extent possible, using the more completely reported variable “Final Energy|Industry|xx” (where xx = Solids, Liquids, or Gases; reported by 87-93%), although this variable can include non-fossil fuel sources. Figures S23 and S24 show the individual, median, and interquartile range (IQR) 2010-2100 pathways, as well as boxplot distributions of the cumulative 2020-2100 values, of the “Final Energy|Non-Energy Use|xx” and “Final Energy|Industry|xx” variables, respectively, as modelled by the C1-C3 scenarios.”

The new methods the authors have introduced to the paper (e.g. the ANOVA, and gas cluster analyses) are beyond my area of expertise. I am unable to comment on whether they are appropriate to use and/or have been carried out correctly processes, although the results make intuitive sense.

As mentioned before, my main concern with the paper is that the main analysis in the paper is quite simple and results are broadly to be expected (they have also been broadly covered in other recent papers e.g. <https://www.nature.com/articles/s41558-022-01576-2>). It therefore remains an editorial decision as to whether this warrants publication.

We think that our paper provides a valuable and timely contribution to the literature and has considerable policy relevance especially in the lead-up to the UN Climate Ambition Summit in September and COP28, as no other paper has yet provided a focused analysis and summary on what the new AR6-assessed mitigation scenarios say about fossil fuel reduction pathways and why, especially with respect to the divergent fossil gas pathways and their implications. The paper cited by the reviewer, on which one of us is also a co-author, explores a related but different issue – looking at the implications of what it would mean if coal cannot be phased out at the rates modelled in the AR6-assessed scenarios given real-world social, economic, and political constraints.

#####

REVIEWERS' COMMENTS

Reviewer #2 (Remarks to the Author):

Thank you for responding to the comments. Happy to see the article published.